# Evaluation of Machine Learning Models for Smart Grid Parameters: Performance Analysis of ARIMA and Bi-LSTM

**Yuanhua Chen** [1,2]**, Muhammad Shoaib Bhutta** [2,*]**, Muhammad Abubakar** [3,*]**, Dingtian Xiao** [2]**,
Fahad M. Almasoudi** [4]**, Hamad Naeem** [5] **and Muhammad Faheem** [6]

[1] College of Mechanical and Vehicle Engineering, Hunan University, Changsha 410082, China;
chenyuanhua202209@163.com
[2] School of Automobile Engineering, Guilin University of Aerospace Technology, Guilin 541004, China;
18487356130@163.com
[3] Key Laboratory of Smart Grid of Ministry of Education, Tianjin University, Tianjin 300072, China
[4] Department of Electrical Engineering, Faculty of Engineering, University of Tabuk,
Tabuk 47913, Saudi Arabia; falmasoudi@ut.edu.sa
[5] Department of Computer Science, King Faisal University, Hofuf 31982, Saudi Arabia;
hamadnaeemh@yahoo.com
[6] Department of Computing Technology and Innovations, University of Vaasa, 65200 Vaasa, Finland;
muhammad.faheem@uwasa.fi
[*] Correspondence: shoaib@guat.edu.cn (M.S.B.); mr_abubakar58@hotmail.com (M.A.)

**Abstract:** The integration of renewable energy resources into smart grids has become increasingly important to address the challenges of managing and forecasting energy production in the fourth energy revolution. To this end, artificial intelligence (AI) has emerged as a powerful tool for improving energy production control and management. This study investigates the application of machine learning techniques, specifically ARIMA (auto-regressive integrated moving average) and Bi-LSTM (bidirectional long short-term memory) models, for predicting solar power production for the next year. Using one year of real-time solar power production data, this study trains and tests these models on performance measures such as mean absolute error (MAE) and root mean squared error (RMSE). The results demonstrate that the Bi-LSTM (bidirectional long short-term memory) model outperforms the ARIMA (auto-regressive integrated moving average) model in terms of accuracy and is able to successfully identify intricate patterns and long-term relationships in the real-time-series data. The findings suggest that machine learning techniques can optimize the integration of renewable energy resources into smart grids, leading to more efficient and sustainable power systems.

**Keywords:** renewable energy; smart grids; energy forecasting; ARIMA; Bi-LSTM model

## 1. Introduction

As the world is moving towards sustainability, there is an increasing need for renewable and green energy resources. Non-renewable resources such as fossil fuels are rapidly disappearing, and their consumption is causing severe environmental hazards. Therefore, renewable energy resources such as solar energy, wind energy, and hydroelectric power are becoming more and more popular [1]. However, the integration of these resources into the smart grid is a challenge that needs to be addressed. Almost all researchers focus on enhancing stability and control in power grid systems through the proposal of hybrid solutions, emphasizing the importance of addressing challenges associated with grid faults, and utilizing specific control strategies to improve fault ride-through capabilities, voltage regulation, and reactive power injection, supported by simulation results and comparisons, while also suggesting future directions for further research [2–5]. In the near future, smart grids such as wind, solar, biogas, and so on will need to be controlled through artificial intelligence techniques to enhance their efficiency [6,7]. The authors of [8] present a novel

approach using the YOLOv5 algorithm and image processing techniques to monitor photovoltaic modules in solar power plants, emphasizing the use of drones and automated detection while suggesting potential enhancements such as thermal infrared integration and robotic cleaning operations. Among renewable energy resources, solar energy has emerged as one of the most important contributors to fulfilling energy demands [9]. Solar plants generate energy on a daily basis, and this energy is provided to national grids. Accurately predicting the energy production of solar plants is a difficult task that requires advanced techniques [10]. Traditional methods are utilized to predict power generation and other parameters, but their accuracy is still limited [10]. To improve the accuracy of these predictions, many researchers have received help from different machine learning models that are implemented on the dataset acquired from different energy and power plants. These machine learning models have proven to be very effective for forecasting/predicting different parameters [11]. In [12], only a single statistical ARIMA model is used for the prediction of solar plant parameters, but the results are not clearly understandable, whereas Ref. [13] proposes a hybrid model of ARIMA and LSTM for short-time electric energy estimation of photovoltaic (PV) power plants. The model combines the advantages of both methods to enhance the estimation accuracy of electric power forecasting. However, the study only evaluates the performance of the model on a single PV plant, which may limit its generalizability to other plants. In [14], a Bi-LSTM model is used only for electricity generation parameter forecasting for a solar PV plant. The model takes into account both temporal and spatial dependencies of input features to improve the accuracy of forecasting, but the study only considers one input feature (i.e., solar irradiance), which may limit the model's ability to capture other factors that affect electricity generation. Similarly, the work of [15] proposes an LSTM neural network for forecasting the electricity generation of a solar PV power plant. This research study executes the comparison of the performance of the LSTM model with other methods, such as ARIMA and random forest. The study only uses a limited amount of data (i.e., one year), which may not be sufficient to capture the full range of variability in electricity generation. Again, the work of [16] compares the performance of ARIMA and LSTM models for forecasting electricity generation from a solar power plant, in which the LSTM model outperforms the ARIMA model in terms of accuracy, but here, the data size and plant capacity are very small, which may limit the generalizability of the findings. A hybrid ARIMA–LSTM model for short-term prediction of PV power output is used in [17]. The model combines the strengths of both methods to improve the accuracy of forecasting, but, owing to limited data availability, it may not be sufficient to capture the full range of variability in PV power output. These machine learning models are also utilized in other kinds of power generation plant datasets, as Ref. [18] compares the performance of ARIMA, LSTM, and ELM models for short-term wind power forecasting. It can be shown that the ELM technique performs better in terms of accuracy than the ARIMA and LSTM methods. The only limitation of this study is that it does not provide a comparison of the computational complexity or feasibility of the different models. In the same way, both of these models are compared with the RF model, SVR model, MLP model, and LSTM-ATT again for wind power and solar power forecasting, and these models outperform the ARIMA and LSTM models in terms of accuracy and stability. However, a lack of explanation of how the models were trained and preprocessed is observed here, which may limit the reproducibility and reliability of the results [19–22]. Additionally, these studies do not account for external factors that may affect the performance of the models, such as changes in weather patterns or policy changes affecting the energy industry. The authors of [23] proposed a Bi-LSTM model for short-term wind power forecasting that incorporates wavelet decomposition and clustering analysis to enhance the accuracy of predictions. The model was found to outperform other models in terms of accuracy, but may not take into account extreme weather conditions, which can cause significant changes in wind power generation. Moreover, the model uses many parameters and does not compare the proposed model to other forecasting methods, which makes it difficult to assess the model's effectiveness and leads to over fitting. In the same way, a Bi-LSTM model

is used with an attention mechanism for electricity load forecasting in [24] to highlight the relevant features in the input data and improve the model's accuracy, as well as in [25] to improve the capturing capacity with an attention mechanism for short-term photovoltaic power forecasting's ability to capture complex patterns in data. Although the proposed model outperforms other models in terms of accuracy and robustness, it only focuses on electricity load forecasting for one region and may not be applicable to other regions with different load patterns [26]. Furthermore, Bi-LSTM is used for short-term solar power forecasting for wind power generation based on multi-step ahead optimization and has been found to outperform other models in terms of accuracy, stability, and robustness [27].

From the above discussion, it is concluded that the only potential limitation of the Bi-LSTM model is that it may be computationally expensive and require a large amount of data for training. The performance of the model may be impacted by the quality and availability of the input data. Therefore, it is important to carefully evaluate the performance of the Bi-LSTM model and consider the limitations and potential drawbacks of these models in energy forecasting applications. This research also focuses on one year of real-time solar power production data and evaluates the performance parameters of the ARIMA and Bi-LSTM models in predicting solar power production and radiance for the following year. Further research and development in this area are needed to continuously improve the accuracy and robustness of renewable energy generation prediction models to support efficient energy management and achieve sustainable energy systems.

Section 2 of the study presents the framework and methodology of the proposed models, including the mathematical models, structure, and a basic comparison of the two models, while Section 3 focuses on the case study and data description. The results of the study, including losses and predictions, are thoroughly discussed in Section 4. Section 5 is the detail of system and software used for the experimental results. Finally, Section 6 presents the conclusive remarks of the study.

## 2. Proposed Framework/Methodology

The methodology for this research paper involves the implementation of an ARIMA (auto-regressive integrated moving average) and a Bi-LSTM (bidirectional long short-term memory) machine learning model on a stationary time series dataset. This dataset consists of one-year values (January 2022 to December 2022) of three parameters i.e., grid-connected power generation (MW), daily generation (kWh), and radiance (MJ·m$^{-2}$). These real-time data are collected from a large-scale solar power plant; further details about this large-scale solar power plant can be found in [13]. The dataset has been pre-processed to ensure that it meets the requirements of a stationary time series. Both of the models are then trained on a specified portion of the dataset, with the remaining dataset being split into testing and validation sets. The models are then evaluated for their performance on the testing set and tuned to optimize accuracy. The validation set is then used to verify the performance on unseen data. Once the models are optimized and validated, they are used to predict power generation for the next year. This is carried out by extrapolating the time series data into the future and forecasting future values of the dependent variable based on the predictions. Overall, the methodology for this research paper involves using rigorous statistical and machine learning methods to analyze a large-scale solar power plant's power production data. Both the ARIMA (auto-regressive integrated moving average) and Bi-LSTM (bidirectional long short-term memory) models are used to perform predictions of future power generation, which might be helpful in making decisions concerning resource allocation, pricing strategies, optimal power system planning, system stability, and other important design parameters for solar power plant operators. Figure 1 demonstrates the complete flowchart of the whole process for predicting the visualization results from both models. The following sub-sections present the structural and mathematical details of the ARIMA (auto-regressive integrated moving average) and Bi-LSTM (bidirectional long short-term memory) models.

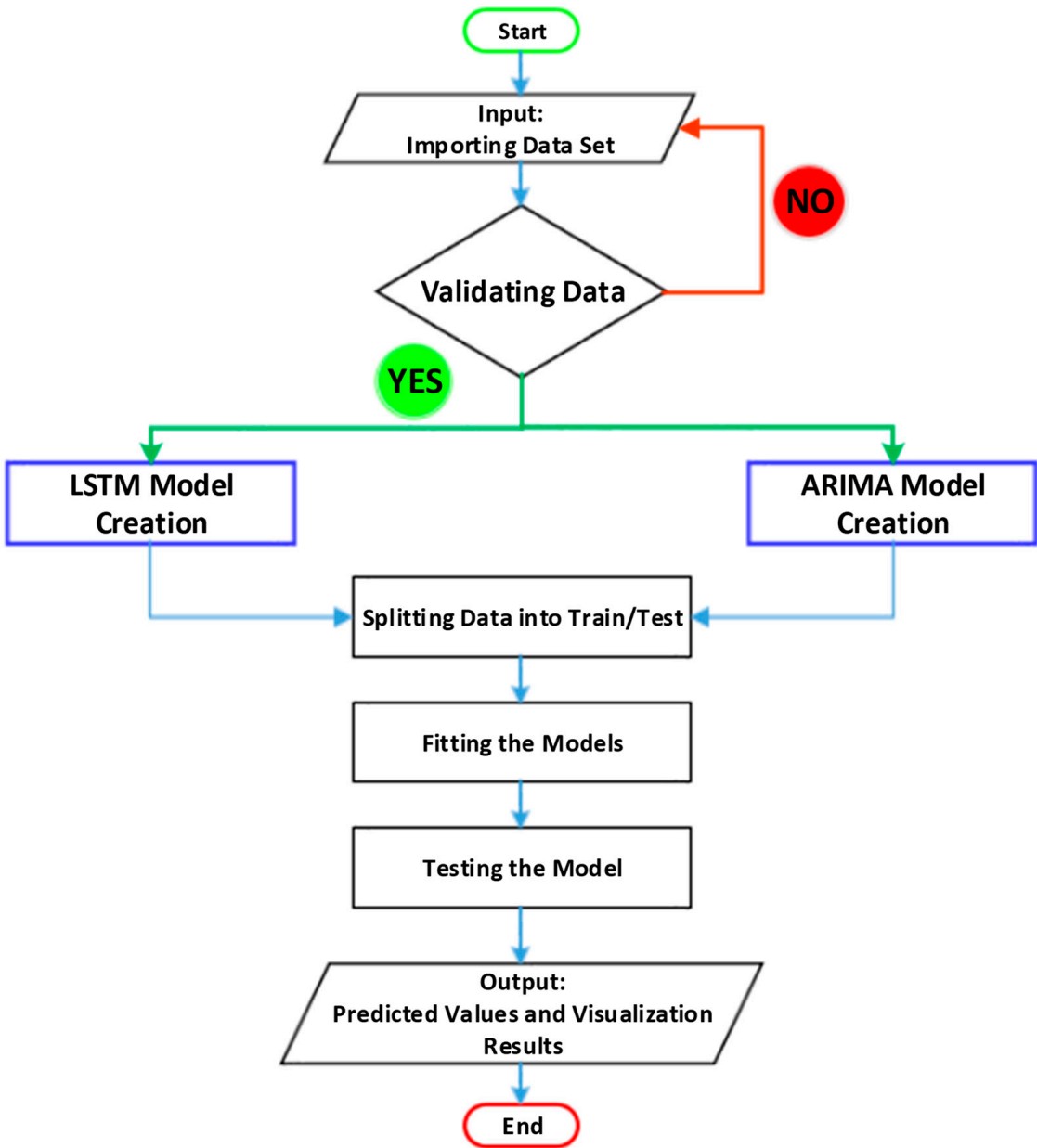

**Figure 1.** Process for visualization results through ARIMA and Bi-LSTM.

### 2.1. Structure of ARIMA Model

The ARIMA (auto-regressive integrated moving average) model is a popular statistical model used for time series forecasting. It consists of three main components, as shown in Figure 2. The first component is auto-regression (AR), which describes the link between an observation and a predetermined number of lag observations. This component is modelled using the observations from previous time periods. The AR component of an ARIMA model is denoted as AR (p), where "p" represents the number of lagged observations to be incorporated. The next component is moving average (MA), which creates a model for the connection that exists between an observation and a predetermined number of lagging prediction errors (i.e., the difference between the actual observation and the forecasted value). The MA component of an ARIMA model is denoted as MA (q), where "q" represents the number of lagged forecast errors to be involved. The third component is integration (I), which models the degree of differencing required to make the time series stationary. The "I" component of an ARIMA model is denoted as I(d), where "d" represents the number of times the time series needs to be differenced to achieve stationarity. Therefore, an ARIMA

model is typically denoted as ARIMA (auto-regressive integrated moving average), where "p" is the order of the autoregressive component, "d" is the degree of differencing required, and "q" is the order of the moving average component.

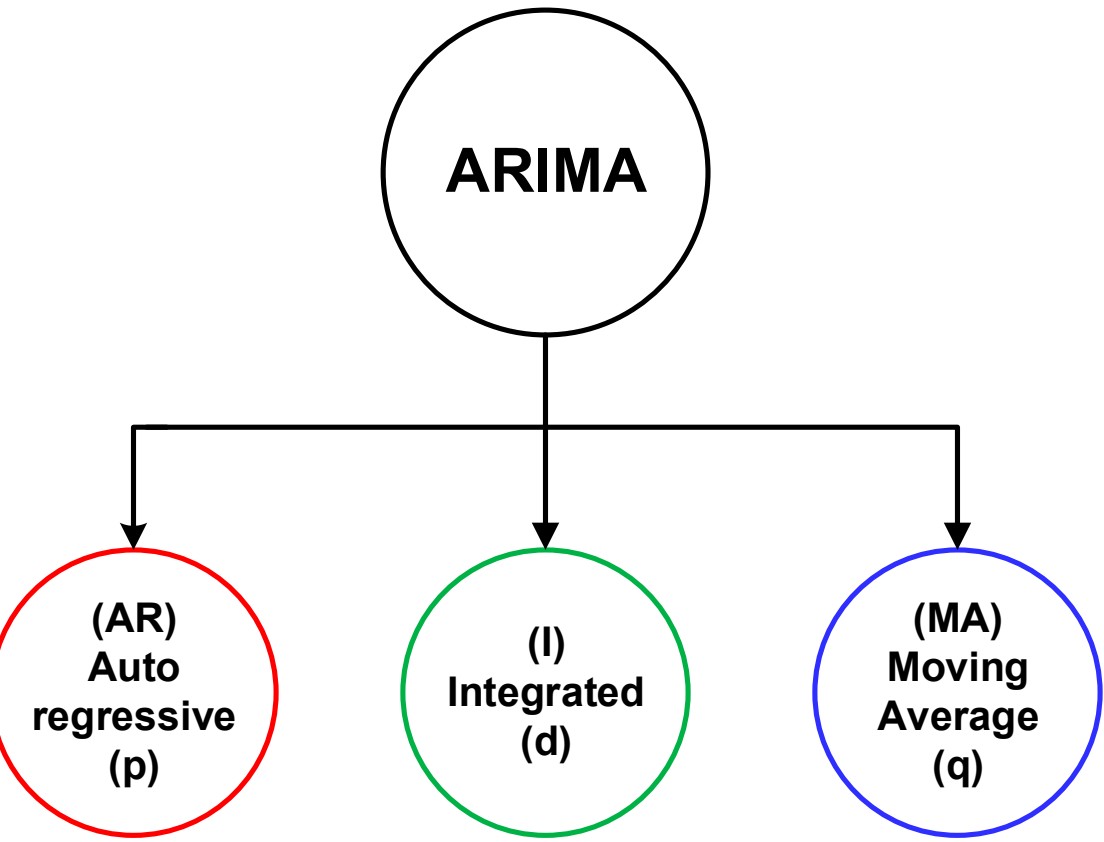

**Figure 2.** Explanation of ARIMA model's abbreviations.

In addition, the ARIMA (auto-regressive integrated moving average) model, which has been used for prediction throughout this investigation, is shown as a forecasting process in Figure 3, which may be found in this case. The flowchart provides an overview of the processes involved in estimating the accuracy of real-time data obtained from a solar plant. The first step is to gather all of the data, which can then be used to make accurate predictions. Training the model, which entails differentiating the series in order to make it stationary, is the next step. The next phase, which occurs after the model has been trained, is to identify the model that will be used to estimate the parameters of the taught model. This step happens after the model has been trained. Following the successful identification of the model, the following stage is to make an estimate of the model's parameters. Following the estimation of the model's parameters, a diagnostic is performed to establish whether or not the model is sufficient. In the event that the model does not adequately represent the data, the procedure will begin once again at the step of model identification. In the event that the model is satisfactory, we may proceed to the forecasting step.

In summary, the flowchart presents a sequential process that involves collecting real-time data, forecasting errors, training the model, identifying the model, estimating its parameters, diagnosing the model, and proceeding to forecasting or prediction. This process allows the accurate prediction of errors in real-time data collected from a solar plant.

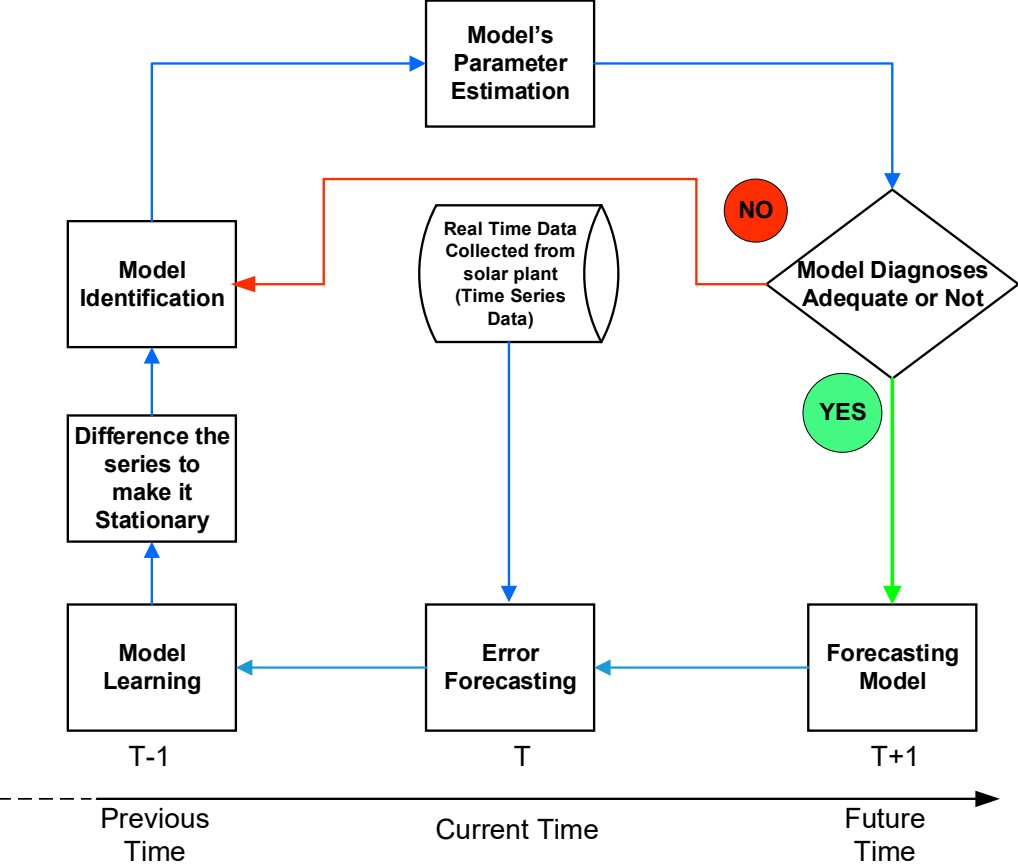

**Figure 3.** Forecasting/prediction process of the ARIMA model.

*2.2. Mathematical Description of the ARIMA Model*

In the ARIMA (auto-regressive integrated moving average) model, Equation (1) shows the current value of the time series, which is a linear function of its previous value, with an intercept term $\beta_0$ and a slope coefficient $\beta_1$ that determines the strength of the autoregressive effect [13]. The error term $\varepsilon_t$ captures the randomness and variability of the time series that cannot be explained by the autoregressive model.

$$Y_t = \beta_1 y_{t-1} + \beta_2 y_{t-2} + \ldots + \beta_0 y_0 + \varepsilon_t \tag{1}$$

where $Y_t$ is the value of the time series at time t. $\beta_0$, $\beta_1$, $\beta_2$, $\ldots$ , $\beta_q$ are the parameters of the autoregressive (AR) component of the model. These parameters describe how much the p values of the time series in the past have influenced the value being measured at the moment. $\beta_0$ represents the intercept term. $Y_{t-1}$, $Y_{t-2}$, $\ldots$ ,$Y_{t-q}$ are the lagged values of the time series up to order p. $\varepsilon_t$ is the error term or residual at time t, which captures the unpredictable, random fluctuations of the time series that are not explained by the AR component. d is the differencing order, which specifies how many times the time series is differenced to make it stationary.

Equation (2) implies that the time series value at $t-1$ is predicted based on its lagged values up to order p and the error term at time $t-1$. In other words, the present value of a time series is contingent on its past results, with a decreasing impact as the lag order increases [13]. The coefficients $\beta_1$, $\beta_2$, $\ldots$ , $\beta_q$ represent the strength of the autoregressive effect for each lag order, while $\beta_0$ represents the baseline level of the time series.

$$Y_{t-1} = \beta_1 y_{t-2} + \beta_2 y_{t-3} + \ldots + \beta_0 y_0 + \varepsilon_{t-1} \tag{2}$$

where $Y_{t-1}$ is the lagged value of the time series at time $t-1$. $\beta_0$, $\beta_1$, $\beta_2$, $\ldots$ , $\beta_q$ are the parameters of the AR component of the model, which define the degree to which prior p

values of a time series impact the present value. $\beta_0$ represents the intercept term. $Y_{t-2}$, $Y_{t-3}, \ldots, Y_{t-q}$ are the lagged values of the time series up to order p. $\varepsilon_{t-1}$ is the error term or residual at time t − 1, which captures the unpredictable, random fluctuations of the time series that are not explained by the AR component.

Equation (3) provides the prediction of the present values of a time series depending on its historical values, weighted by the AR model coefficients [13]. The error term represents the difference between the actual value and the predicted value and is used to estimate the model's accuracy and improve its forecasting performance.

$$Y_t = \beta_1 y_{t-1} + \beta_2 y_{t-2} + \ldots + \beta_0 y_0 + \varepsilon_t \tag{3}$$

where $Y_t$ is the value of the time series at time t. $Y_{t-1}, Y_{t-2}, \ldots, Y_{t-q}$ are the past values of the time series up to time 0. $\beta_1, \beta_1, \beta_2, \ldots$ are the coefficients of the ARIMA (autoregressive integrated moving average) model. They represent the weights given to the previous values of the time series in order to predict its current value. The error term $\varepsilon_t$ or innovation represents the differentiation of actual values existing in the time series at 't' and its relevant predicted values based on the previous values.

Equation (4) represents the moving average (MA) component of the ARIMA (autoregressive integrated moving average) model, which is utilized to estimate the current values of a time series determined by its previous error value [13].

$$Y_t = \alpha + \varepsilon_t + \varnothing_1 \varepsilon_{t-1} + \varnothing_2 \varepsilon_{t-2} + \ldots + \varnothing_q \varepsilon_{t-q} \tag{4}$$

where $Y_t$ is the value of the time series at time t. $\alpha$ is the intercept or constant term of the model, which represents the expected value of the time series when all other variables are equal to 0. $\varepsilon_t$ denotes the differentiation of the actual values of the time series at time t and its relevant predicted values based on the past values and the previous errors. $\varnothing_1, \varnothing_2, \ldots$, $\varnothing_q$ are the ARIMA model's moving average (MA) coefficients. They represent the weights given to the previous errors in order to predict the current value of the time series. $\varepsilon_{t-1}$, $\varepsilon_{t-2}, \ldots, \varepsilon_{t-q}$ are the previous errors, which are the differences between the actual and predicted values of the time series at the previous time steps.

Equation (5) represents the combination of both the AR and MA components in the ARIMA (auto-regressive integrated moving average) model [13]. It is used to forecast the current value of a time series based on its past values and prior errors, with weights provided to each value and error depending on the coefficients of the AR and MA models. The current value of a time series is predicted based on its past values and previous errors. The anticipated value of the time series is represented by the intercept term when all other variables are set to zero, and the error term is the difference between the actual value and the projected value at the current time step. Both terms are referred to collectively as the error term.

$$Y_t = \alpha + \beta_1 y_{t-1} + \beta_2 y_{t-2} + \beta_p y_{t-p} \varepsilon_t + \varnothing_1 \varepsilon_{t-1} + \varnothing_2 \varepsilon_{t-2} \ldots + \varnothing_q \varepsilon_{t-q} \tag{5}$$

where $Y_t$ is time series value at t. The model's intercept, or constant term, is the time series' anticipated value when all other variables are zero. $\beta_1, \beta_2, \ldots, \beta_q$ are the coefficients of the autoregressive (AR) component of the ARIMA (auto-regressive integrated moving average) model. They are a representation of the weights that have been assigned to the previous values of the time series in order to make a prediction about the current value. $Y_{t-1}, Y_{t-2}, \ldots$, $Y_{t-q}$ are the lagged values of the time series at time t − 1, t − 2, . . . , t − p. $\varepsilon_t$ indicates the differentiation of the actual values of the time series at time t and its relevant predicted values based on the previous values and the previous errors. $\varnothing_1, \varnothing_2, \ldots, \varnothing_q$ are the ARIMA model's moving average (MA) coefficients. They represent the weights given to the previous errors in order to predict the current value of the time series. $\varepsilon_{t-1}, \varepsilon_{t-2}, \ldots$, $\varepsilon_{t-q}$ are the previous errors, which are the differences between the actual and predicted values of the time series at the previous time steps.

### 2.3. Structure of Bi-LSTM

The bidirectional long short-term memory (Bi-LSTM) model is a sort of recurrent neural network (RNN) that is frequently used for time series forecasting, sequence classification, and other sequential data processing applications. The Bi-LSTM (bidirectional long short-term memory) model has a more complex structure than the ARIMA (auto-regressive integrated moving average) model. The basic structure of the bidirectional long short-term memory model is shown in Figure 4. In this model, sequential input data are brought into the Bi-LSTM (bidirectional long short-term memory) model via the first layer, which is referred to as the input layer, as mentioned in Figure 4. The second layer is known as the bidirectional LSTM layer and it is made up of two different sets of LSTM units. The input sequence is handled by the first set in the conventional forward direction, while the second set handles it in the conventional reverse manner. The model is able to properly represent both past and future dependencies in the input sequence because of this feature. The dropout or output layer is the third layer, which prevents over fitting by dropping out part of the units from the bidirectional LSTM layer in a random fashion while the layer is being trained. The fourth layer is known as the dense layer and it is responsible for producing the final output by taking the output from the LSTM layer and applying a linear transformation to it. In addition, activation functions and regularization may be added to the output using this layer, which can also be utilized. Therefore, the Bi-LSTM (bidirectional long short-term memory) model may be layered with numerous layers to expand its capacity and capture more complicated patterns in the input sequence. This can be accomplished by stacking the models. Additionally, the model can be trained using back propagation through time (BPTT) to update the model parameters based on the sequence of past inputs and outputs, as shown in Figure 4. The model parameters, including the number of LSTM units, number of layers, and dropout rate, can be optimized using techniques such as grid search optimization.

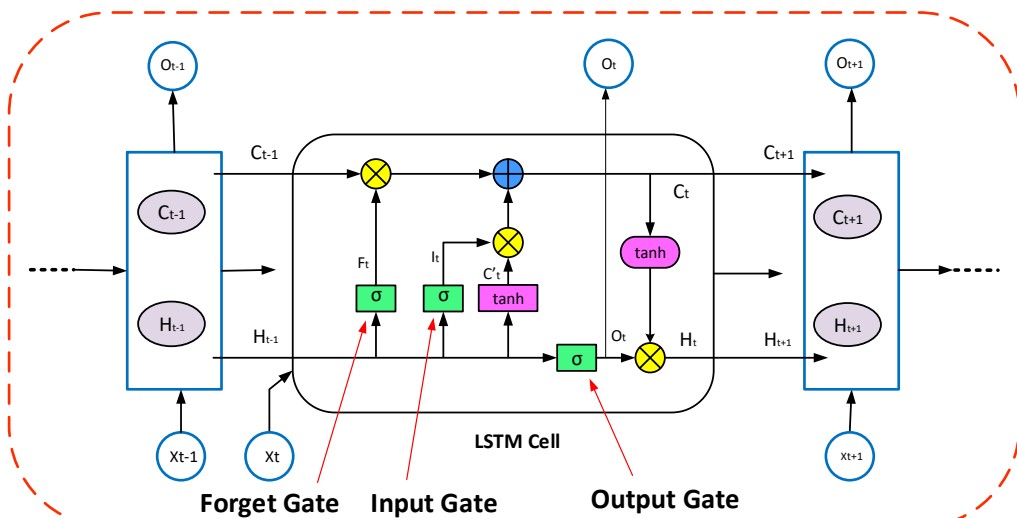

**Figure 4.** Bi-LSTM internal structure.

### 2.4. Mathematical Explanation of Bi-LSTM

Equation (6) describes the operation of the input gate in a bidirectional long short-term memory (Bi-LSTM) model, which computes a weighted sum of the input $x_t$, the previous hidden state $h_{t-1}$, and a bias term $b_i$. These three terms are first multiplied by the learned weight matrices $W_i$ and $U_i$ and then added together. In the end, the sigmoid function is used to calculate the value of the input gate by applying itself to the total [28].

$$i_t = \sigma(W_i[x_t] + U_i[h_{t-1}] + b_i) \tag{6}$$

where $i_t$ is the input gate vector at time step t and $x_t$ is the input vector at time step t, whereas h(t − 1) is hidden state at time step t − 1. Wi and weight matrix for the input gate are applied to input $x_t$ and hidden state h(t − 1), respectively. Bi is bias term for the input gate. Equation (7) describes the operation of the forget gate in a bidirectional long short-term memory (Bi-LSTM) model, which computes a weighted sum of the input $x_t$, the previous hidden state hidden state $h_{t−1}$, and a bias term $b_f$. These three terms are first multiplied by the learned weight matrices $W_f$ and $U_f$ and then added together. Finally, the sigmoid function is used on the total before the value of the forget gate can be determined $f_t$ [29].

$$f_t = \sigma(W_f[x_t] + U_f[h_{t−1}] + b_f) \tag{7}$$

where $f_t$ is the forget gate vector and $x_t$ is the input vector at time step t. $h_{t−1}$ is the hidden state at time step t − 1, whereas $W_f$ is the weight matrix for the forget gate applied to input $x_t$ and $U_f$ is the weight matrix for the forget gate applied to the hidden state $h_{t−1}$. $b_o$ is the bias term for the forget gate. Equation (8) shows the calculation of the candidate memory cell state in a bi-directional long short-term memory (Bi-LSTM) network [30]. Once the weighted sums and bias are added together, the resulting vector is passed through the hyperbolic tangent function (tanh) to produce the candidate memory cell state $c'_t$. The hyperbolic tangent function compresses the values of the input vector to the range $[−1, 1]$, which helps to prevent the exploding gradient problem that can occur during training.

$$c'_t = \tan h(W_a[x_t] + U_a[h_{t−1}] + b_a) \tag{8}$$

where $c'_t$ is the candidate activation vector and $x_t$ is the input vector at time step t. $h_{t−1}$ is the hidden state at time step t − 1. $W_a$ is the weight matrix for candidate activation applied to input $x_t$ and $U_a$ is the weight matrix for candidate activation applied to the hidden state $h_{t−1}$. $b_o$ is the bias term for candidate activation. Equation (9) represents the calculation for the cell state of a single time step t in a bidirectional long short-term memory (Bi-LSTM) model [31]. The forget gate $f_t$ decides which information from the previous cell state should be kept and which should be discarded. If the forget gate outputs $f_t$= 0, then all of the information from the previous cell state is forgotten, and if $f_t$ = 1, all information is retained. Then, the new information is added to the cell state through the input gate and determines which parts of the candidate cell state $c'(t)$ should be stored in the cell state. If $i_t$ = 0, no new information is added, and if $i_t$ = 1, all of the new information is stored. Finally, the two parts are added together to update the cell state $c_t$ for the current time step t.

$$c_t = (f_t \times c_{t−1}) + i_t \times c'_t \tag{9}$$

where $c_t$ is the cell state vector and $c_{t−1}$ is the cell state vector at time step t and t − 1, respectively. $f_t$ is the forget gate vector and $i_t$ is the input gate vector at time step t. $c'_t$ is the candidate activation vector at time step t.

Equation (10) computes the output state $o_t$ at time step t. It is a function of the current input $x_t$; the previous hidden state $h_{t−1}$; and learnable parameters $W_o$, $U_o$, and $b_o$. These parameters are learned during training and are used to weight and combine the input and previous hidden state to compute the output state [31].

$$o_t = \sigma(W_o[x_t] + U_o[h_{t−1}] + b_o \tag{10}$$

where $o_t$ is the output gate vector and $x_t$ is the input vector at time step t, $h_{t−1}$ is the hidden state at time step t − 1, $W_o$ and $U_o$ are the weight matrix for the output gate applied to input $x_t$ and the hidden state $h_{t−1}$, and $b_o$ is the bias term for the output gate. Equation (11) computes the hidden state $h_t$ at time step t. It is a function of the output state $o_t$ and the memory cell state $C^t$. The memory cell state is computed based on the previous hidden

state $h_{t-1}$ and the current input $x_t$. The output state is then computed based on the current hidden state $h_t$ and used to selectively filter information from the memory cell [31].

$$h_t = (o_t) \times \tan h(C^t) \tag{11}$$

where $h_t$ represents the hidden state, $o_t$ is the output gate, and $c_t$ is the cell state vector at time 't'.

### 2.5. Comparison between ARIMA and Bi-LSTM

ARIMA (auto-regressive integrated moving average) and Bi-LSTM (bidirectional long short-term memory) are both time series forecasting models, but they differ in approach and assumptions. ARIMA (auto-regressive integrated moving average) is a statistical model that assumes stationary data, while Bi-LSTM (bidirectional long short-term memory) is a deep learning model that can handle both stationary and non-stationary data, as well as varying lengths and missing values. Bi-LSTM (bidirectional long short-term memory) has shown advantages over ARIMA (auto-regressive integrated moving average) in terms of accuracy and flexibility, particularly for complex and dynamic data.

### 3. Case Study

Box plots are useful for summarizing a large dataset and providing insights into its distribution. Figure 5 shows the median (Q2), interquartile range (Q1–Q3), and outliers or extreme values of the parameters of data acquired from a large-scale solar power plant, including grid-connected power generation in MW, daily generation in kWh, and radiance in MJ·m$^{-2}$. The height of the box represents the spread of the middle 50% of the data, and the whiskers extend to the lowest and highest values that fall within 1.5 times the interquartile range. Any values outside this range are shown as individual points or circles, and can be considered potential outliers. The first box plot is "grid-connected power generation in MW", and the *y*-axis depicts the range of electric power generation (20 MW–100 MW). The range of Q1 (the first quartile) is 58–68, which implies that the lowest 25% of the data falls in this range. The range of Q2 (the second quartile) is 68–73, which indicates that the middle 50% of the data falls in this range. The range of Q3 (the third quartile) is 73–78, which signifies that the highest 25% of the data falls in this range. Finally, the range of Q4 (the fourth quartile) is 78–99, which suggests that the top 1% of the data falls in this range. In the same way, the second box plot is "daily generation in kWh", and the *y*-axis depicts the range of number of kWh units generated (100,000 kWh–600,000 kWh). The range of Q1 is 260,000–420,000, which presents that the lowest 25% of the data falls in this range. The range of Q2 is 420,000–470,000, which demonstrates that the middle 50% of the data falls in this range. The range of Q3 is 470,000–510,000, which represents that the highest 25% of the data falls in this range. Finally, the range of Q4 is 510,000–570,000, which signifies that the top 1% of the data falls in this range. The third box plot is "radiance in MJ·m$^{-2}$", and the *y*-axis represents the range of effective solar radiation (5 MJ·m$^{-2}$–30 MJ·m$^{-2}$) that has the potential to be converted into electrical energy. The range of Q1 is 10–18, which signifies that the lowest 25% of the data falls in this range. The range of Q2 is 18–22, which presents that the middle 50% of the data falls in this range. The range of Q3 is 22–24, which indicates that the highest 25% of the data falls in this range. Finally, the range of Q4 is 24–27, which implies that the top 1% of the data falls in this range, as shown in Figure 5 below.

Figure 6 represents the correlation of the data through a heat map. A heat map is a visual representation of data that uses color-coded cells to show the relative values of different variables. In this case, the heat map shows the correlation between three variables, grid-connected power generation in MW, daily generation in kWh, and radiance in MJ·m$^{-2}$. The numbers shown in each cell are indicative of the correlation coefficient that exists between the two variables associated with the row and column that the cell represents.

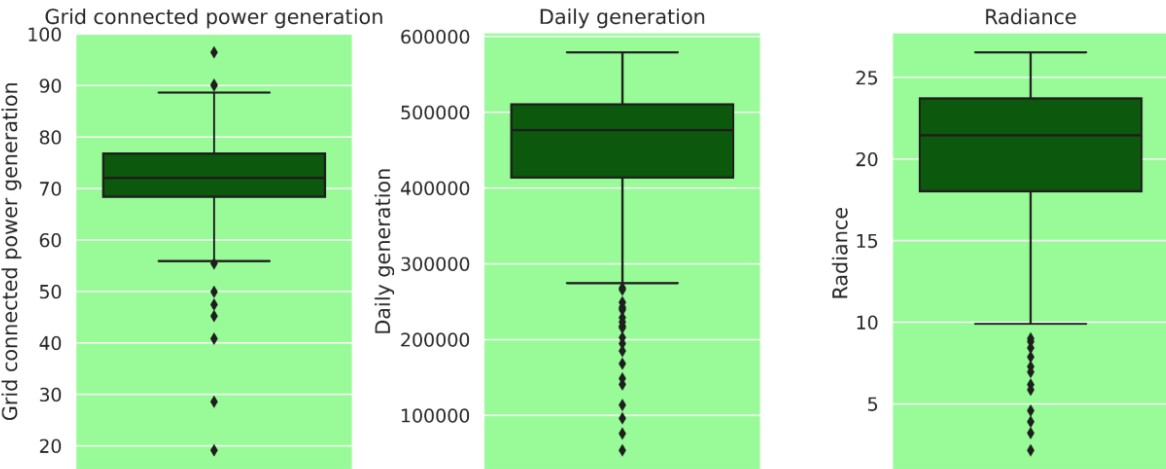

**Figure 5.** Box plots of data parameters of the solar power generation plant.

**Figure 6.** Heat map showing the correlation between the data parameters of the solar power generation plant.

The correlation coefficient ranges from −1 to +1, with values close to −1 indicating a strong negative correlation, values close to +1 indicating a strong positive correlation, and values close to 0 indicating little or no correlation. The heat map presented has three rows and three columns, corresponding to the three variables. The diagonal cells (i.e., the cells where the row and column variables are the same) are all 1, because a variable is always perfectly correlated with itself. The off-diagonal cells show the correlation between pairs of variables. For example, the cell in the first row and second column shows the correlation between grid-connected power generation in MW and daily generation in kWh, which is 0.37. This indicates a positive but relatively weak correlation between the two variables. Similarly, the cell in the second row and third column shows the correlation between daily generation (kWh) and radiance (MJ·m$^{-2}$), which is 0.95. This indicates a strong positive correlation between the two variables. Overall, the heat map shows that there is some correlation between the three variables, but the strength of the correlations varies. Grid-connected power generation in MW and daily generation in MJ·m$^{-2}$ are weakly correlated, while daily generation (kWh) and radiance (MJ·m$^{-2}$) are strongly correlated.

A histogram visualizes the frequency distribution of the dataset, in which the *x*-axis represents the range of values in the dataset, while the *y*-axis represents the frequency of those values. In Figure 7, the *y*-axis measures the frequency of occurrence of grid-connected power generation values from 0.00 to 0.06, likely in units of megawatts (MW).

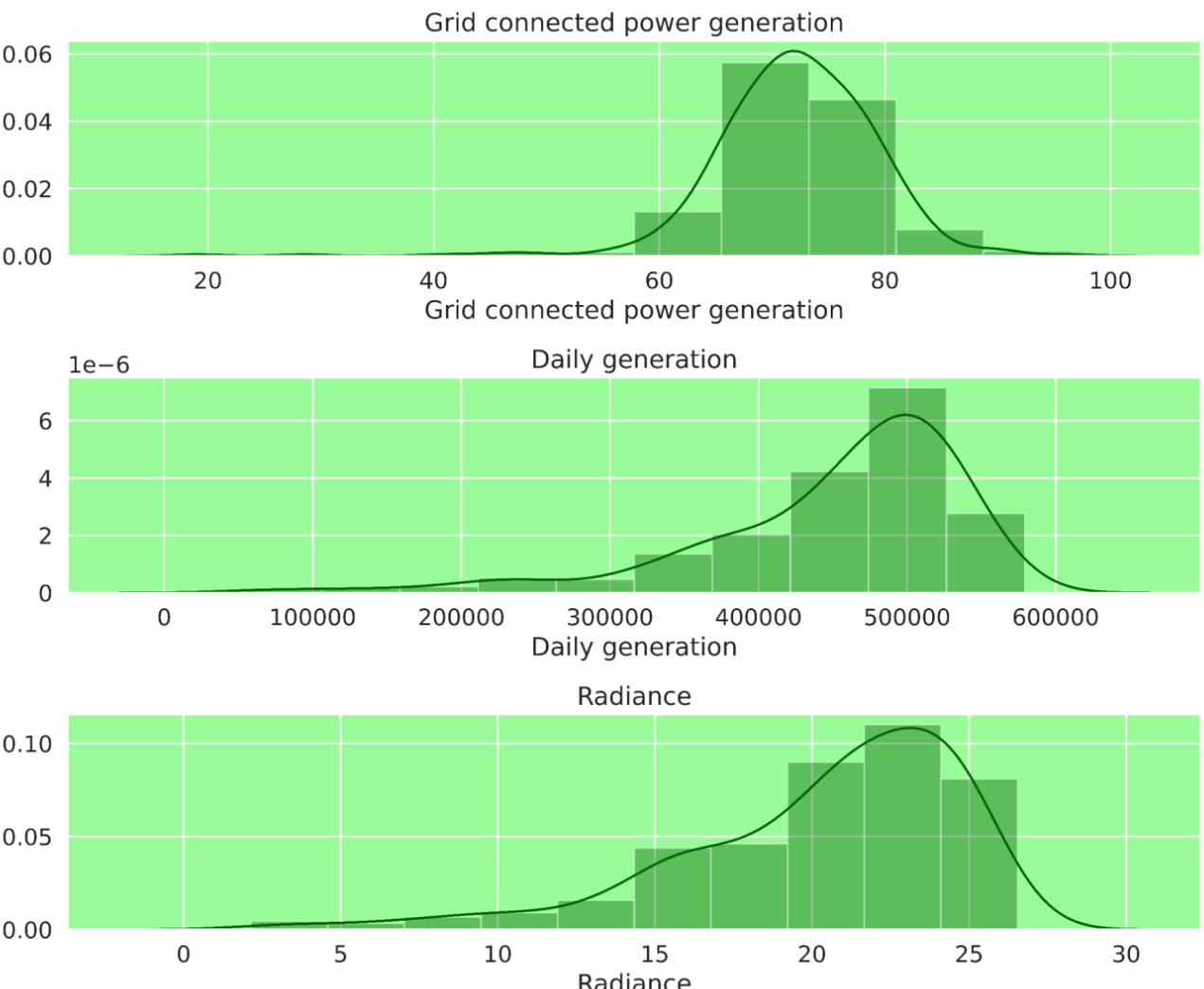

**Figure 7.** Histogram visualization of the frequency distribution of the dataset of a large-scale solar power generation plant.

The *x*-axis measures grid-connected power generation values from 0 to 100 MW and is divided into equal bins to create the histogram. By counting the occurrences of grid-connected power generation values within each bin and plotting the frequencies on the *y*-axis, the resulting histogram shows the distribution of power generation values over a specific period of time. The second part of histogram, titled "daily generation (kWh)", represents the distribution of the number of units generated by a solar power plant on a daily basis. The *y*-axis range is 0.00–7.00, indicating the frequency of occurrence of the number of units generated. The *x*-axis range is 0–600,000 units, divided into equal intervals or bins, representing the range of values of the number of units generated. Similarly, the "radiance" histogram has a *y*-axis range of 0.00–0.10 and an *x*-axis range of 0–30. The *y*-axis shows the frequency or count of data in each interval, with the highest count being 0.10 and the lowest count being 0.00. The *x*-axis shows the range of radiance values grouped into intervals of width 1, with the last interval being 29–30. The histogram represents the distribution of radiance values, where the shape of the histogram can provide insights into the distribution of radiance values. If the histogram is skewed to the right, there are more low radiance values, and if it is skewed to the left, there are more high radiance values. A symmetrical histogram indicates an even distribution of radiance values.

## 4. Results and Discussion

In this study, a one-year dataset is collected from a solar plant, capturing grid-connected power generation (MW), daily generation (kWh), and radiance (MJ·m$^{-2}$) parameters in real time. These three parameters are utilized to forecast the following year's grid-connected power generation, daily generation, and radiance. This prediction is accomplished using two distinct models: the autoregressive integrated moving average (ARIMA) and a machine learning bidirectional long short-term memory (Bi-LSTM) model. The selection of these models was based on a comprehensive literature review, which identified them as the most accurate and refined approaches for this purpose. While the results of both models indicate predictions that are very close to the actual real-time data values, the machine learning Bi-LSTM (bidirectional long short-term memory) model outperforms the ARIMA (auto-regressive integrated moving average) model. In this section, we discuss the results and graphical visualizations of both models, along with detailed analysis of the mean absolute error and mean square error values.

The first analysis is conducted using the ARIMA (auto-regressive integrated moving average) model, which is a statistical approach to prediction. For the purpose, data analysis was carried out between actual and predicted data through four statistical plots that are commonly used in data analysis, as shown in Figures 8–10. These figures contain four graphical visualizations. The first one is the standardized distribution plot, which standardizes a variable for comparison to other variables that may have different scales or units. The second is a histogram that visualizes the distribution of a variable, identifying its shape and any outliers. The third is a normal Q–Q plot, which assesses whether a sample distribution is normal. Finally, the forth is a correlogram, which helps to identify correlations between a variable and its lagged values and is also useful in time series analysis. Each plot serves a specific purpose in data analysis and can provide valuable insights into the underlying patterns and trends of the data.

Figure 8a is the standardized residual graph, which is a plot of the residuals against the standardized values of the independent variable of the grid-connected power generation data of the solar plant. In the case of grid-connected power generation data, the *y*-axis range is −8 to 6 and the *x*-axis range is 0–350. The first three negative peaks appear at −6, −5.2, and −7, indicating significant deviations from the predicted values. However, after the third peak, the residuals are within the range of −2 to 4 up to day 350, suggesting that the model fits the data well. The standardized residual graph can help identify any outliers or unusual data points that may require further investigation. Figure 8b is a histogram showing the distribution of power generation from a grid-connected solar power plant. The *y*-axis represents the frequency of power generated, which ranges from

0.0 MW to 0.7 MW, while the *x*-axis represents the deviation of power generated from its mean value. The highest peak of the histogram bar indicates that the most frequent power generation lies between 0 and 0.55 MW. The KDE and N (0, 1) density curves estimate the probability density function, and their highest peaks indicate that the most probable power generation lies in the range of 0.6 MW and 0.4 MW, respectively. This graph provides a visual representation of the distribution of power generated by a solar power plant. The normal Q–Q plot in Figure 8c is used to assess the normality of residuals in statistical models, such as the ARIMA (auto-regressive integrated moving average) model for predicting power generation (MW) in a solar power plant. The *y*-axis depicts the observed residuals and the *x*-axis shows the expected residuals based on a normal distribution. The straight red line represents the line of normality and deviation from this line indicates non-normality. In the given plot, the residuals have a positive skew, but more than 90% of the data falls close to the straight red line, indicating that the residuals are approximately normally distributed. The normal Q–Q plot is a useful tool for assessing the accuracy of the ARIMA (auto-regressive integrated moving average) model predictions. Figure 8d is a correlogram, which is used for the autocorrelation function (ACF) of a time series at different lags in ARIMA (auto-regressive integrated moving average) modeling to identify the appropriate order of the model by examining the decay of the ACF. Most of the data points in the correlogram indicate no significant correlation between the time series and its lagged versions, but a few data points deviate from the 0.0 line, which may indicate the presence of some autocorrelation. Further investigation is needed to determine the appropriate ARIMA (auto-regressive integrated moving average) model order.

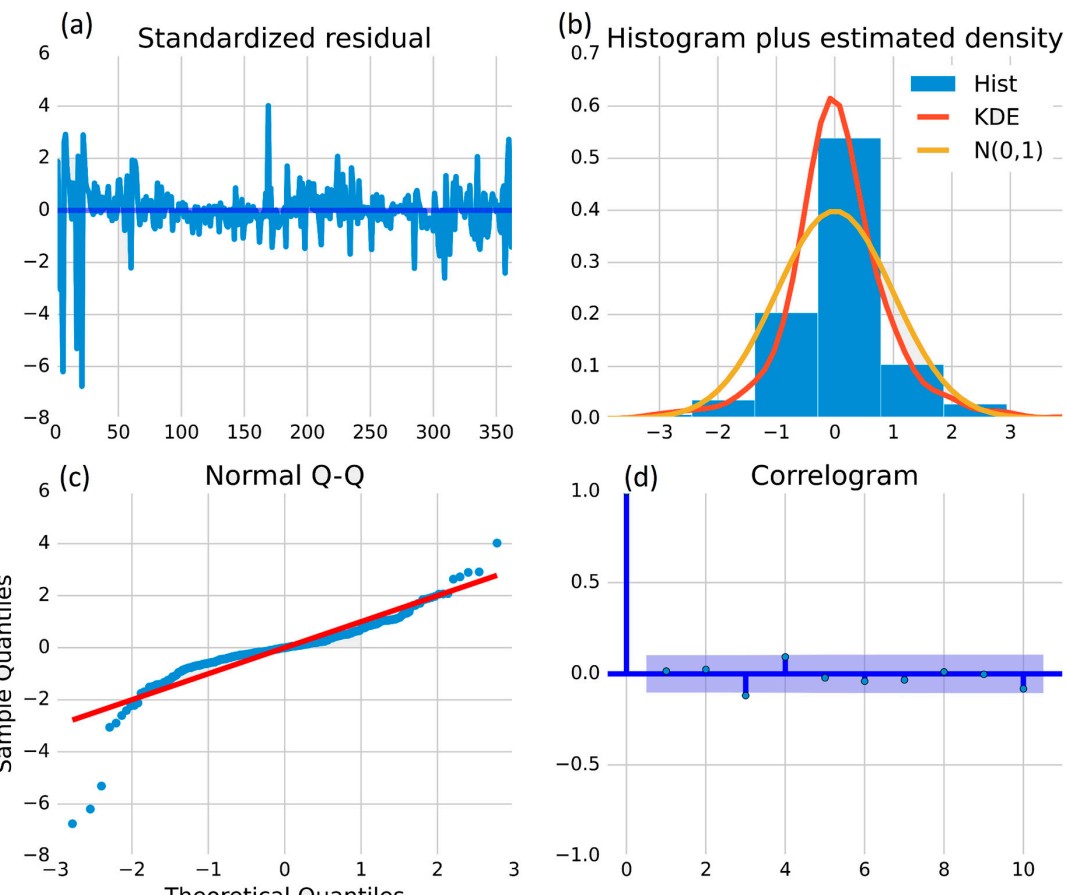

**Figure 8.** (**a**) Standardized distribution plot for grid-connected power generation data, (**b**) distribution visualized through histogram for grid-connected power generation data, (**c**) distribution analysis through normal Q–Q plot for grid-connected power generation data, and (**d**) correlation between variables and lagged values of grid-connected power generation data.

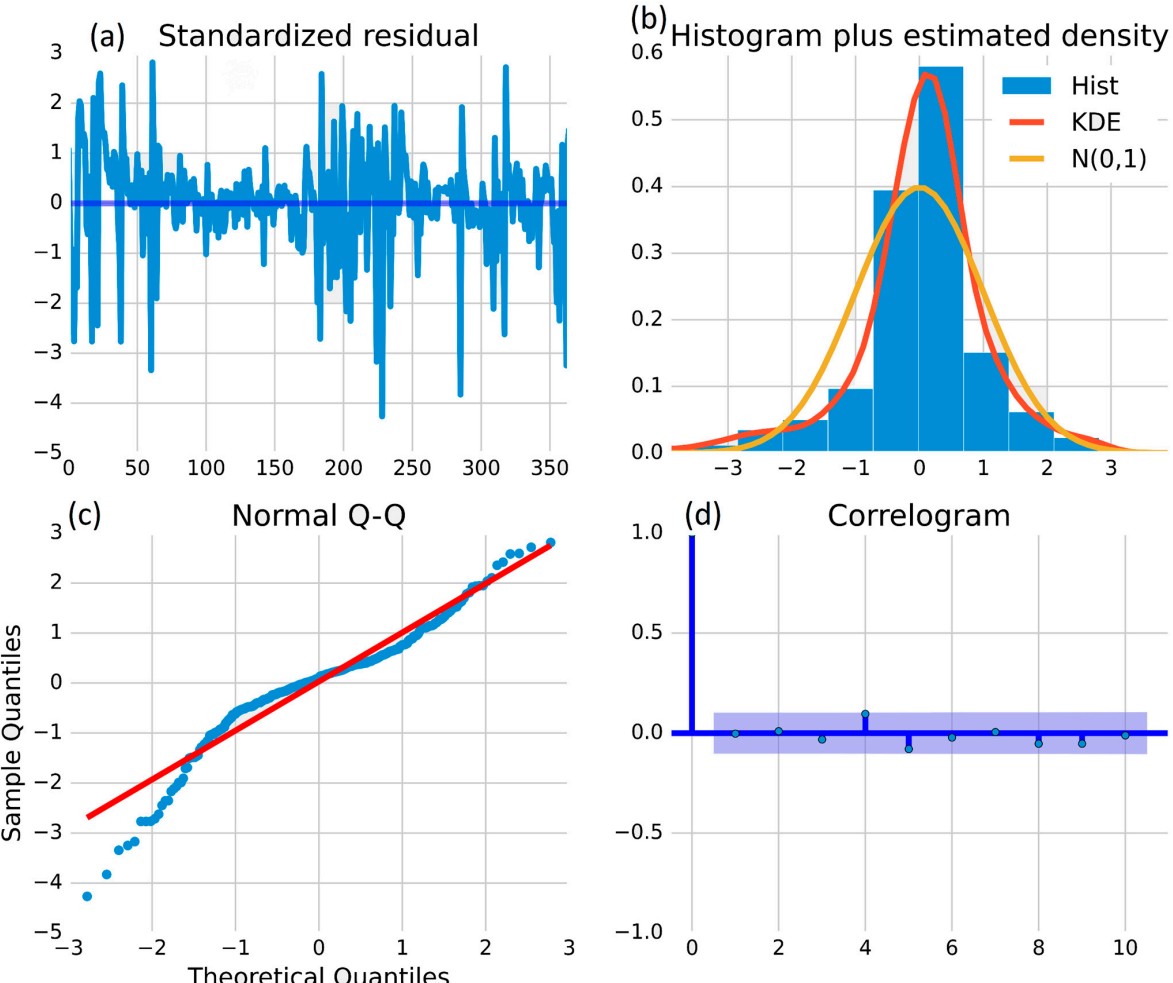

**Figure 9.** (**a**) Standardized distribution plot for daily power generation data of a solar plant, (**b**) distribution visualized through histogram for daily power generation data of a solar plant, (**c**) distribution analysis through normal Q–Q plot of daily power generation data of a solar plant, and (**d**) correlation between variables and lagged values of daily power generation data of a solar plant.

Similarly, Figure 9a is the standardized residual graph plotting the residuals against the standardized values of the independent variable. In the case of a daily solar power plant's daily generation (kWh), the *y*-axis range is −5 to 3 and the *x*-axis range is 0–350. The highest negative peak appears at −4.2 and the highest positive peak appears at 2.9, indicating significant deviations from the predicted values on those days. After the highest peaks, all of the residuals are within the range of −4.2 to 2.8 up to day 350, indicating that the model fits the data well. The standardized residual graph can help identify any outliers or unusual data points that may require further investigation. The graph in Figure 9b represents the distribution of the grid daily generation (kWh) of a solar power plant using a histogram, a kernel density estimation (KDE) plot, and a normal distribution plot. The histogram shows that the highest peak of the histogram bar is nearly equal to 0.6, indicating that the solar power plant generates power close to 0.6 most of the time. The KDE plot shows the probability density function of the data, with the highest positive peak appearing at 0.55 in the red line. The yellow line represents the normal distribution, with the highest positive peak appearing at 0.4. Comparing the distribution of the solar power plant to the normal distribution provides insight into how the solar power plant's generation behaves relative to a standard distribution. The normal Q–Q plot in Figure 9c is a graphical method used to assess the normality assumption of errors in the ARIMA (auto-regressive integrated moving average) model implementation on the real-time data of grid-connected power

generation (MW) of a solar power plant. The *y*-axis represents sample quintiles, the *x*-axis represents theoretical quintiles, and a red line represents the line of perfect agreement. Initially, there is some scattering of data, indicating that the data may not follow a normal distribution. However, from the point (−2.5, −2.5), the data points align more closely with the red line, suggesting that the errors of the ARIMA (auto-regressive integrated moving average) model follow a normal distribution from this point onwards. In the same way, Figure 9d is a correlogram that shows how much each past observation of a time series is related to the current observation. When using the ARIMA (auto-regressive integrated moving average) model on solar power plant data, the correlogram helps to determine the correlation between current and past power generation values. The *y*-axis shows the strength and direction of the correlation, while the *x*-axis represents the number of lagged values being compared. Most data points are near the 0.0 line, but deviations suggest correlation at specific lags and can inform the choice of ARIMA (auto-regressive integrated moving average) model parameters.

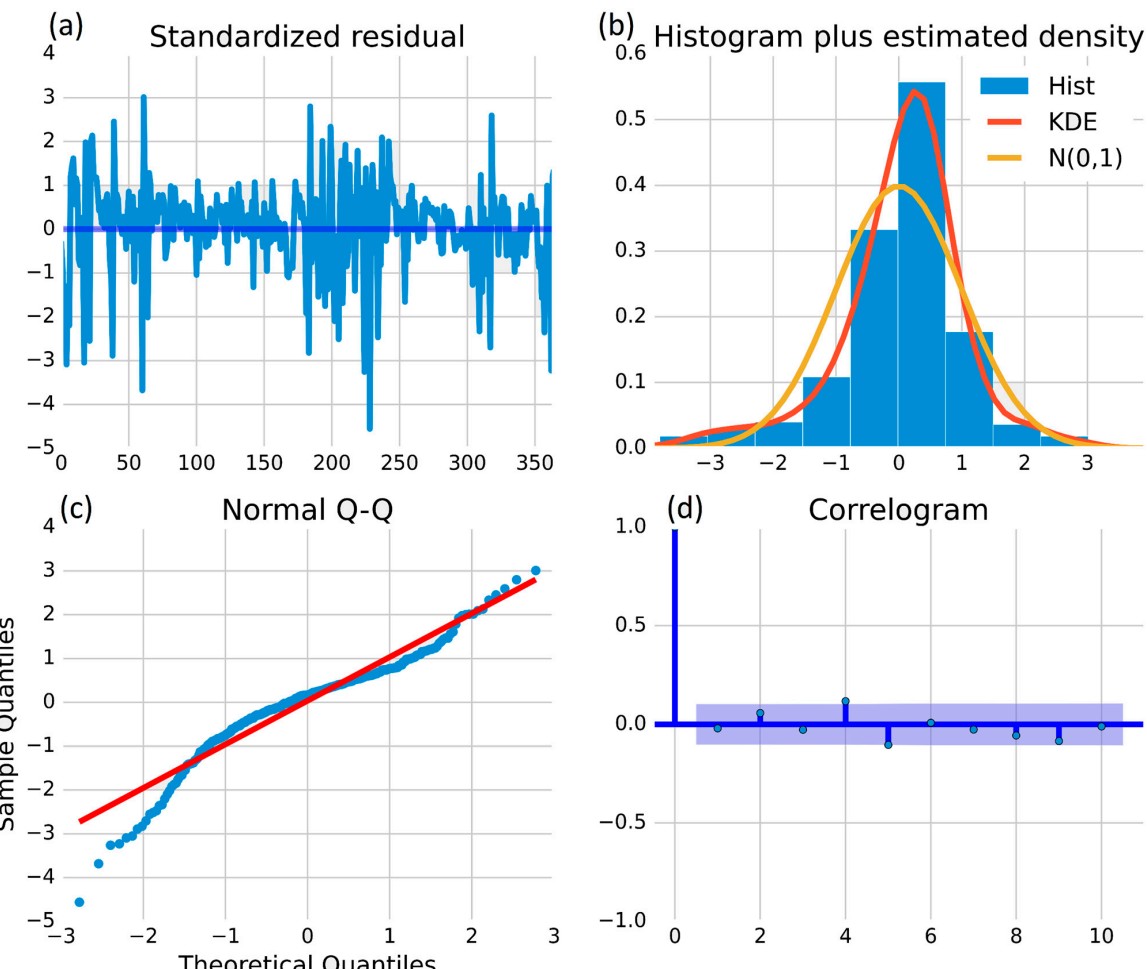

**Figure 10.** (**a**) Standardized distribution plot for radiance data of a solar plant, (**b**) distribution visualized through histogram for radiance data of a solar plant, (**c**) distribution analysis through normal Q–Q plot of radiance data of a solar plant, and (**d**) correlation between variables and lagged values of radiance data of a solar plant.

In the case of a solar power plant, a standard residual graph can be used to predict the radiance (MJ·m$^{-2}$) that the plant will receive based on various factors. In Figure 10a, the *y*-axis ranges from −5 to 4, indicating that the standardized residuals range from −5 standard deviations below the mean to 4 standard deviations above the mean. The *x*-axis ranges from 0 to 350, which represents the predicted radiance values. The fact

that the highest negative peak appears at −4.5 and the highest positive peak appears at 3 suggests that the model is doing a good job of predicting the radiance values. A well-fitted model will have residuals that are randomly distributed around zero with no apparent pattern and will not have any extreme outliers or clusters of outliers. Similarly, in Figure 10b, the graph represents the distribution of the radiance of a solar power plant using a histogram, a kernel density estimation (KDE) plot, and a normal distribution plot. The $x$-axis ranges from −3 to 3, while the $y$-axis ranges from 0.0 to 0.6. The histogram shows that the highest peak of the histogram bar during 0 to 1 is nearly equal to 0.55, indicating that most of the radiance values fall between 0 and 1. The KDE plot shows the probability density function of the data, with the highest positive peak appearing at 0.53 in the red line. This means that the probability density is highest around 0.53, indicating that the radiance values are concentrated around this value. The yellow line represents the normal distribution, with the highest positive peak appearing at 0.4. Comparing the distribution of the radiance of the solar power plant to the normal distribution shows that the radiance values are skewed to the right, with more values falling in the higher range. Overall, the graph provides insight into the distribution of the radiance values of the solar power plant, with the histogram showing the frequency of occurrence of each value and the KDE plot and normal distribution plot showing the probability density function of the data. The normality assumption of errors in the ARIMA (auto-regressive integrated moving average) model implementation on the real-time data of grid-connected power generation of a solar power plant assessed using a normal Q–Q plot is shown in Figure 10c.

The $y$-axis represents sample quantiles, the $x$-axis represents theoretical quantiles, and a red line represents the line of perfect agreement. Initially, there is some scattering of data, indicating that the data may not follow a normal distribution. However, from the point (−2.5, −2.5), the data points align more closely with the red line, suggesting that the errors of the ARIMA (auto-regressive integrated moving average) model follow a normal distribution from this point onwards. In the same way, the correlogram obtained from implementing an ARIMA (auto-regressive integrated moving average) model on real-time-series data of grid-connected power generation (MW) of a solar power plant shows the correlation coefficients between lagged values of the time series in Figure 10d. The $y$-axis range is −1 to 1, with 0 indicating no correlation and −1 and 1 indicating perfect negative and positive correlations, respectively. The $x$-axis range is 0 to 10, representing the number of lagged values being compared to the current value. The data points are mostly clustered around the 0.0 line, indicating little to no correlation between the current power generation and past values, with only a few deviations suggesting some correlation at specific lags. These deviations can be used to inform parameter choices for the ARIMA (auto-regressive integrated moving average) model.

Secondly, the Bi-LSTM (bidirectional long short-term memory) model architecture is a sequential model, with several bidirectional layers followed by dropout and dense layers. The prediction results, MAE loss, and RMSE loss from this model are also discussed in this section and represented in graphical visualizations. Table 1 describes the Bi-LSTM (bidirectional long short-term memory) model's structure, in which bidirectional layers have 200 units each and are stacked on top of each other. The final bidirectional layer outputs a tensor of shape (none, 200), which is then passed through a dropout layer with a rate of 0.5. The output of the dropout layer is then fed to a dense layer with one output unit, which is the final prediction. The summary table provides the output shape and the number of determinable parameters included in every layer of the simulated model. The simulated model has a total of 1,045,001 trainable parameters. The second part of the output shows the keys of the history object returned by the model.fit() method during training. These include the training loss, mean squared error (MSE), and mean absolute error (MAE), as well as the validation loss, MSE, and MAE.

**Table 1.** Bi-LSTM model procedure for prediction.

| Model: "Sequential_7" | | |
|---|---|---|
| **Layer (Type)** | **Output Shape** | **Param #** |
| bidirectional_35 (Bidirectional) | (None, 1, 200) | 81,600 |
| dropout_14 (Dropout) | (None, 1, 200) | 0 |
| bidirectional_36 (Bidirectional) | (None, 1, 200) | 240,800 |
| bidirectional_37 (Bidirectional) | (None, 1, 200) | 240,800 |
| bidirectional_38 (Bidirectional) | (None, 1, 200) | 240,800 |
| bidirectional_39 (Bidirectional) | (None, 200) | 240,800 |
| dropout_15 (Dropout) | (None, 200) | 0 |
| dense_7 (Dense) | (None, 1) | 201 |
| Total params: 1,045,001 Trainable params: 1,045,001 Non-trainable params: 0 | | |
| None dict_keys(['loss', 'mse', 'mae', 'val_loss', 'val_mse', 'val_mae']) | | |

The training and validation results of the Bi-LSTM (bidirectional long short-term memory) model are compared in Figure 11a–c using parameters of data collected from a solar power plant. As can be shown in Table 2, the model achieves the lowest loss on "grid-connected power generation (MW)" at 0.00872, followed by "daily generation (kWh)" at 0.01951, and "radiance (MJ·m$^{-2}$)" at 0.02041. Grid-connected power generation seems to be the most reliably predicted variable in our model. Similar patterns may be seen in the validation loss values, which are used to evaluate the model's adaptability to fresh inputs. Further, the model's validation loss is smallest for the parameter "grid-connected power generation (MW)" at 0.00822, followed by "daily generation (kWh)" at 0.0187 and "radiance (MJ·m$^{-2}$)" at 0.02067. Based on these findings, "grid-linked power generation" seems to be the model's sweet spot in terms of generalizability to fresh data. Prediction accuracy and generalizability to new data are best, as shown by the model's lower validation loss relative to its actual loss. Losses and validation losses for "daily generation (MW)" and "radiance (MJ·m$^{-2}$)" are likewise not too far off the best-performing parameter, indicating that the model performs rather well for these two scenarios.

Table 2 displays the results of a comparison between the training and validation mean absolute errors (MAEs) for a Bi-LSTM (bidirectional long short-term memory) model trained with data from a solar power plant's three major parameters. The MAE loss values for "grid-connected power generation (MW)", "daily generation (kWh)", and "radiance (MJ·m$^{-2}$)" are shown in Figure 12a–c, respectively, and show that the Bi-LSTM (bidirectional long short-term memory) model performs best on "grid-connected power generation (MW)", with a loss of 0.05932. These findings imply that "grid-connected power generation (MW)" is the parameter for which the model's predictions are most accurate. Similarly, the model has a lower mean absolute error (MAE) for "grid-connected power generation (MW)" (0.06027), "daily generation (kWh)" (0.08906), and "radiance (MJ·m$^{-2}$)" (0.0925) than their respective benchmarks. This suggests that, of the three factors, "grid-connected power generation (MW)" benefits most from the model's capacity to generalize to new data. The model's performance is better at predicting this parameter than the other two based on the comparison of MAE loss and validation MAE values.

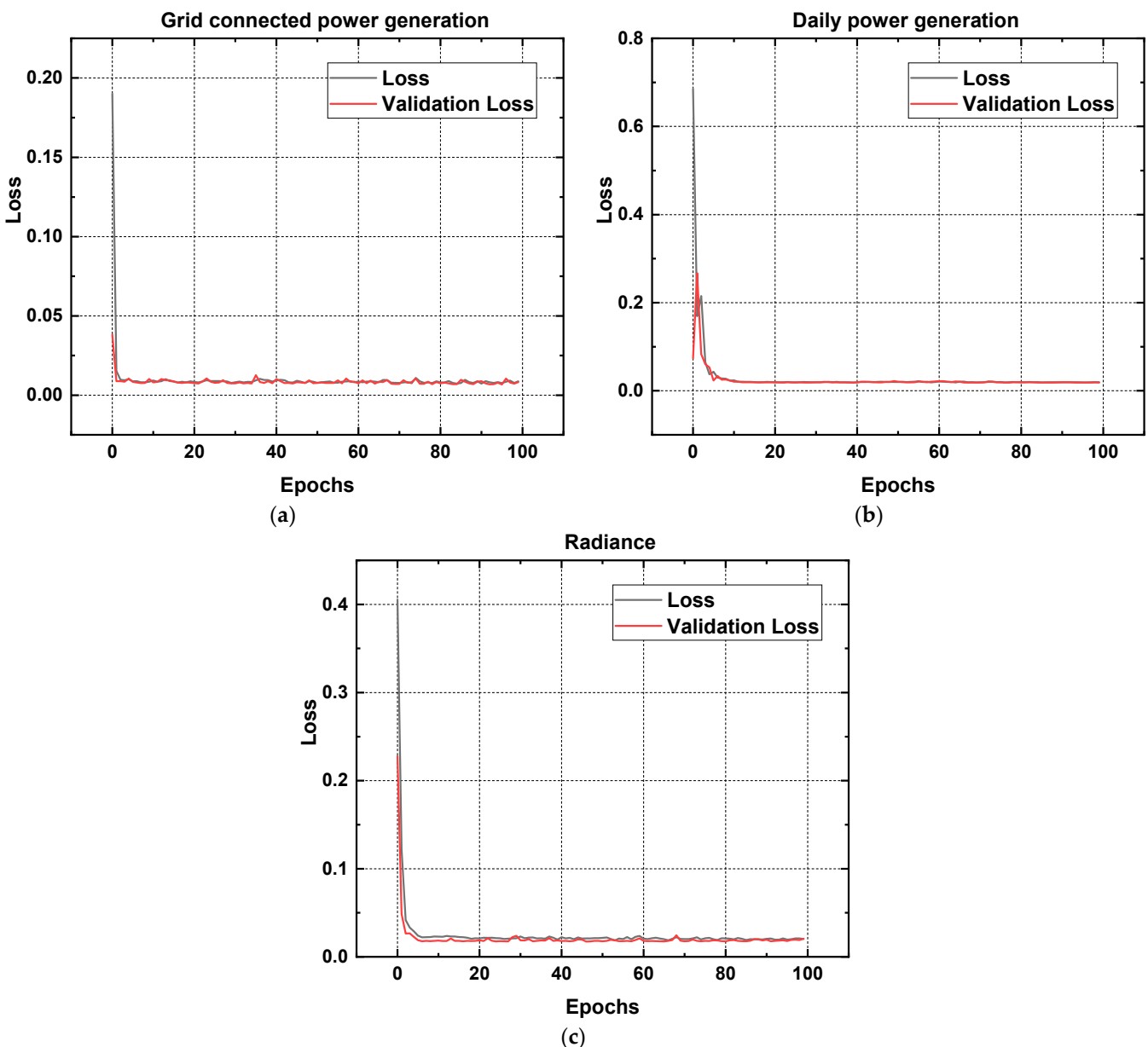

**Figure 11.** (**a**) Actual vs. validation loss of grid-connected power generation of a solar plant; (**b**) actual vs. validation loss of daily power generation of a solar plant; and (**c**) actual vs. validation loss of radiance of a solar plant.

**Table 2.** Performance evaluation of bidirectional LSTM in terms of losses.

| Parameter | Loss | MAE Loss | RMSE Loss | Validation Loss | Validation MAE | Validation MSE |
|---|---|---|---|---|---|---|
| Grid-Connected Power Generation | 0.00872 | 0.05932 | 0.00872 | 0.00822 | 0.06027 | 0.0073 |
| Daily Generation | 0.01951 | 0.09465 | 0.01951 | 0.0187 | 0.08906 | 0.01884 |
| Radiance | 0.02041 | 0.09894 | 0.02041 | 0.02067 | 0.0925 | 0.01912 |

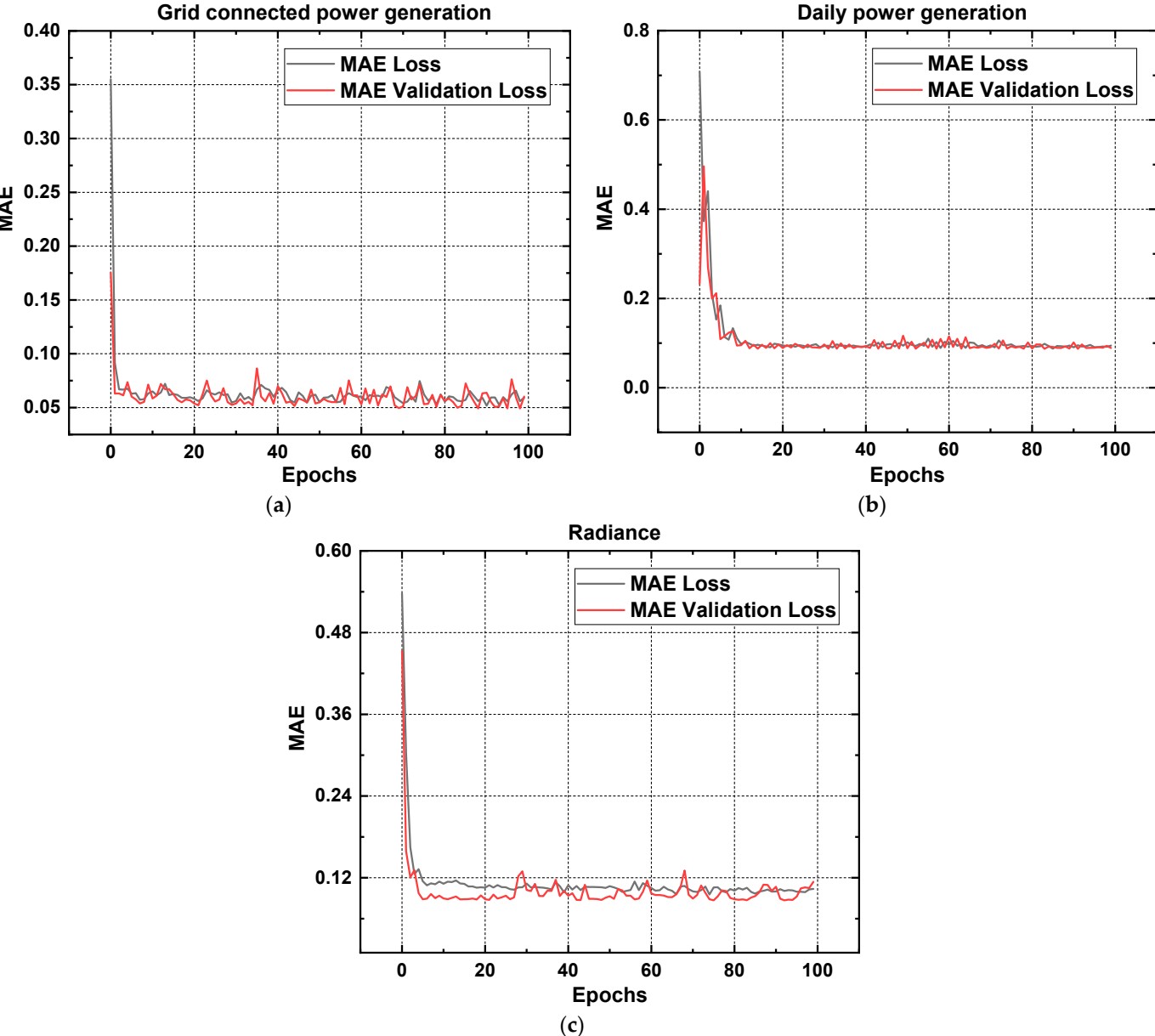

**Figure 12.** (**a**) Actual MAE vs. validation MAE loss of grid-connected power generation of a solar plant; (**b**) actual MAE vs. validation MAE loss of daily power generation of a solar plant; and (**c**) actual MAE vs. validation MAE loss of radiance of a solar plant.

Figure 13a–c compare the root mean squared error (RMSE) loss and validation RMSE values for a Bi-LSTM machine learning model trained on three distinct parameters of a solar power plant. With a validation RMSE of 0.0073 and an RMSE loss of 0.00872 on "grid-connected power generation (MW)", this model outperforms. After "radiance (MJ·m$^{-2}$)", which has an RMSE loss of 0.02041, "daily generation (kWh)" has an RMSE loss of 0.01951 and a validation RMSE of 0.01884. These findings indicate that, among the three factors, "grid-connected power generation (MW)" has the best model predictions in terms of training and generalization to new data. Both the RMSE loss and validation RMSE values for "daily generation (kWh)" and "radiance (MJ·m$^{-2}$)" are similar to the best-performing parameter, indicating that the model's performance in these areas is likewise quite excellent.

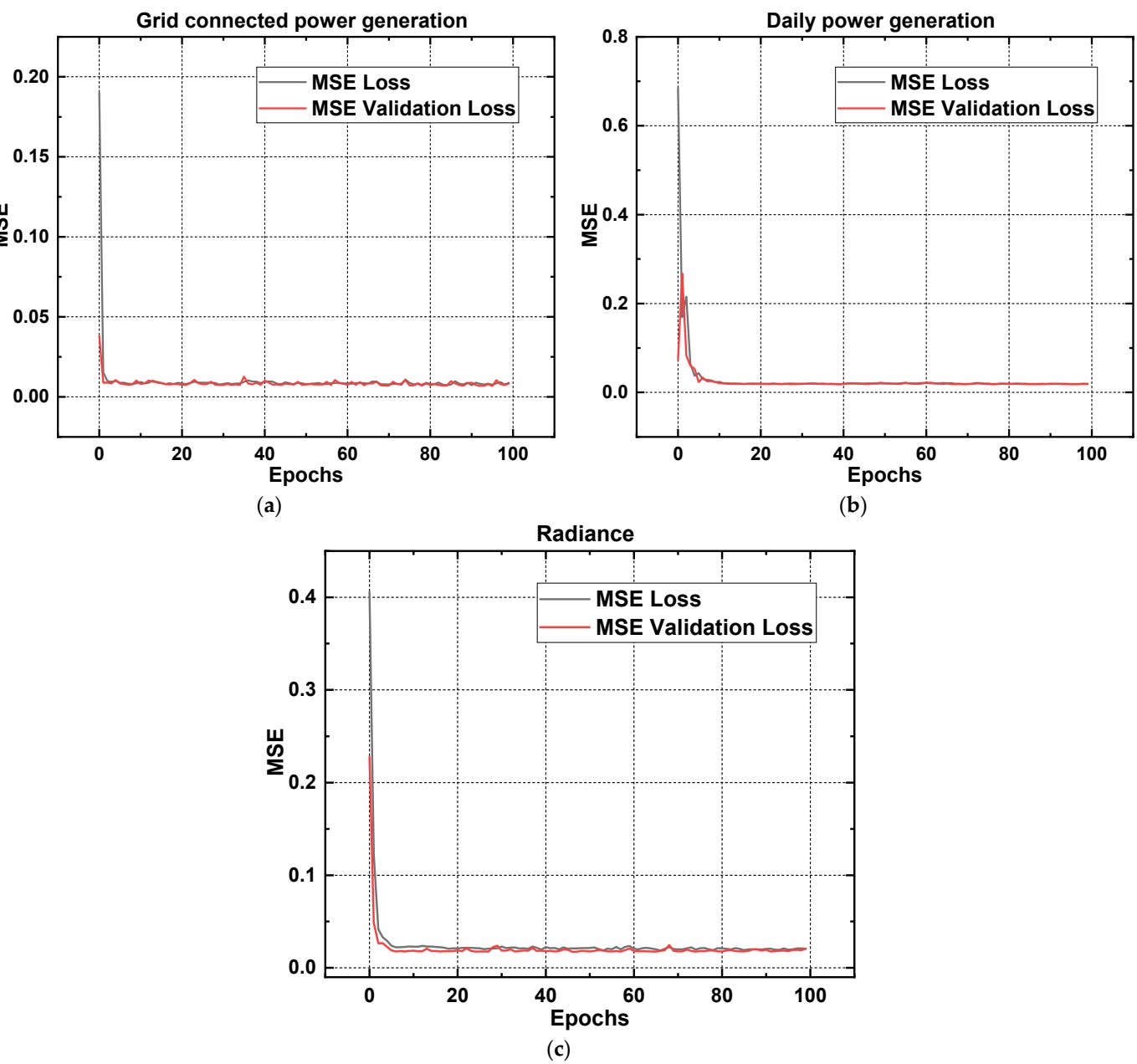

**Figure 13.** (**a**) Actual MSE vs. validation MSE loss of grid-connected power generation of a solar plant; (**b**) actual MSE vs. validation MSE loss of daily power generation of a solar plant. (**c**) Actual vs. validation loss of radiance of a solar plant.

Table 2 presents all of the actual and validated loss values acquired from the Bi-LSTM model implementation on the data of a solar plant.

The prediction outcomes of a data analysis project conducted on a 100 MW solar plant are visually represented in Figures 14–16. This study involved training two machine learning models, namely ARIMA (auto-regressive integrated moving average) and Bi-LSTM (bidirectional long short-term memory), utilizing 80% of one year's real-time data on three crucial parameters: grid-connected power generation (MW), daily power generation (kWh), and radiance (MJ·m$^{-2}$). The remaining 20% of the data was used for testing and validation purposes. The two models' performances were compared by analyzing the 10-month training data against the 2-month test data. Additionally, one-year future predictions were generated for all three parameters using both models. The graphical visualizations in Figures 14–16 provide a comparative analysis of the two models' results

and future predictions for all three parameters within a single graph. The first section of the visualizations depicts the comparison among the actual prediction dataset and the 60-day test dataset of the solar plant. The second part of the graph shows one-year future projections of the solar plant based on each parameter.

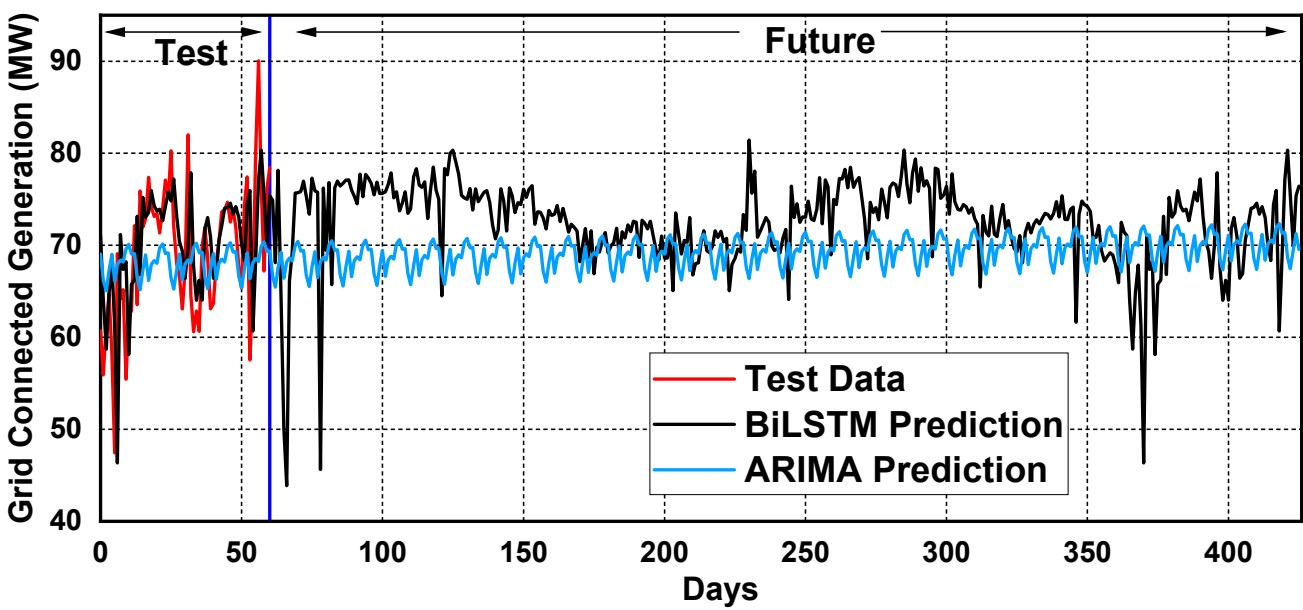

**Figure 14.** Graphical visualization of the comparison between tested and predicted data and one-year prediction of grid-connected power generation of a solar plant.

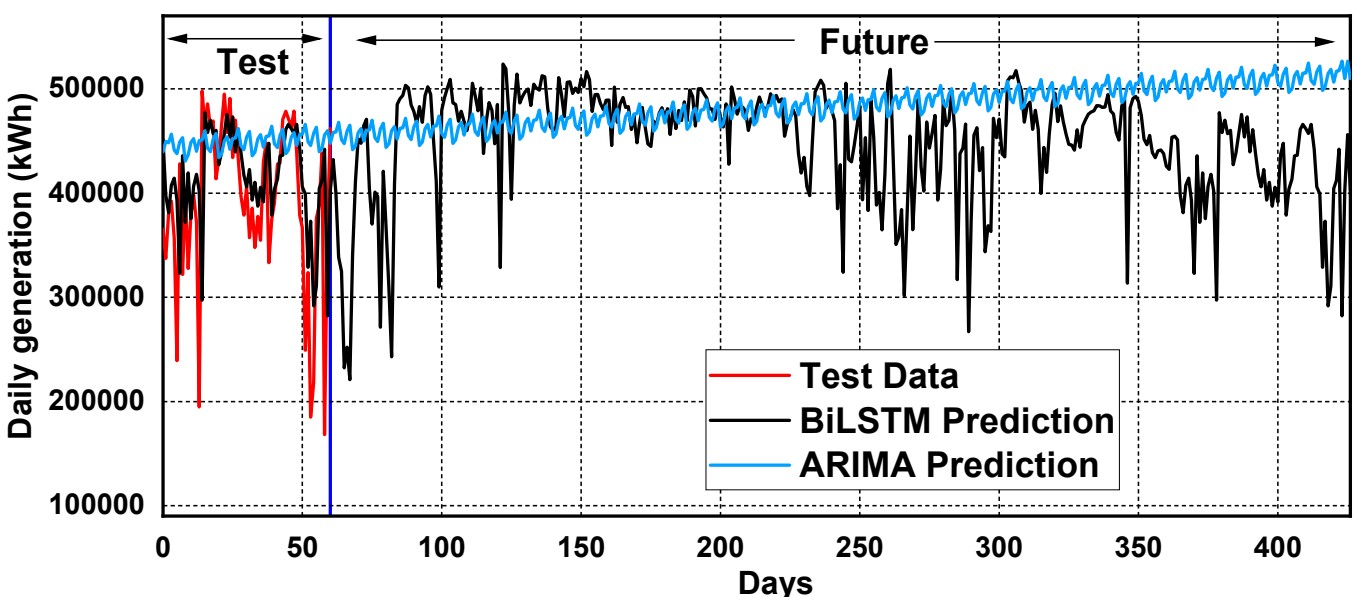

**Figure 15.** Graphical visualization of the comparison between tested and predicted data and one-year prediction of grid daily generation of a solar plant.

Figure 14 displays the "grid-connected power generation (MW)" parameter's range of 40 MW to 100 MW on the *y*-axis, with the number of days on the *x*-axis. The comparison of predicted data using ARIMA (auto-regressive integrated moving average) and Bi-LSTM (bidirectional long short-term memory) and the remaining actual 60-day test data shows that Bi-LSTM's prediction is approximately in sync with the test results, while ARIMA's results have a minor deviation. After this comparison, both trained prediction models are used to manipulate the grid-connected power generation data for the next 12 months

(365 days). It is evident in Figure 14 that Bi-LSTM (bidirectional long short-term memory) initially shows an abrupt decrease and then achieves a continuous pattern, indicating that the grid-connected power generation will remain smooth and upgraded most of the time for the next year. Meanwhile, a slight increase in grid-connected power generation is observed in ARIMA's prediction, with a continuity trend.

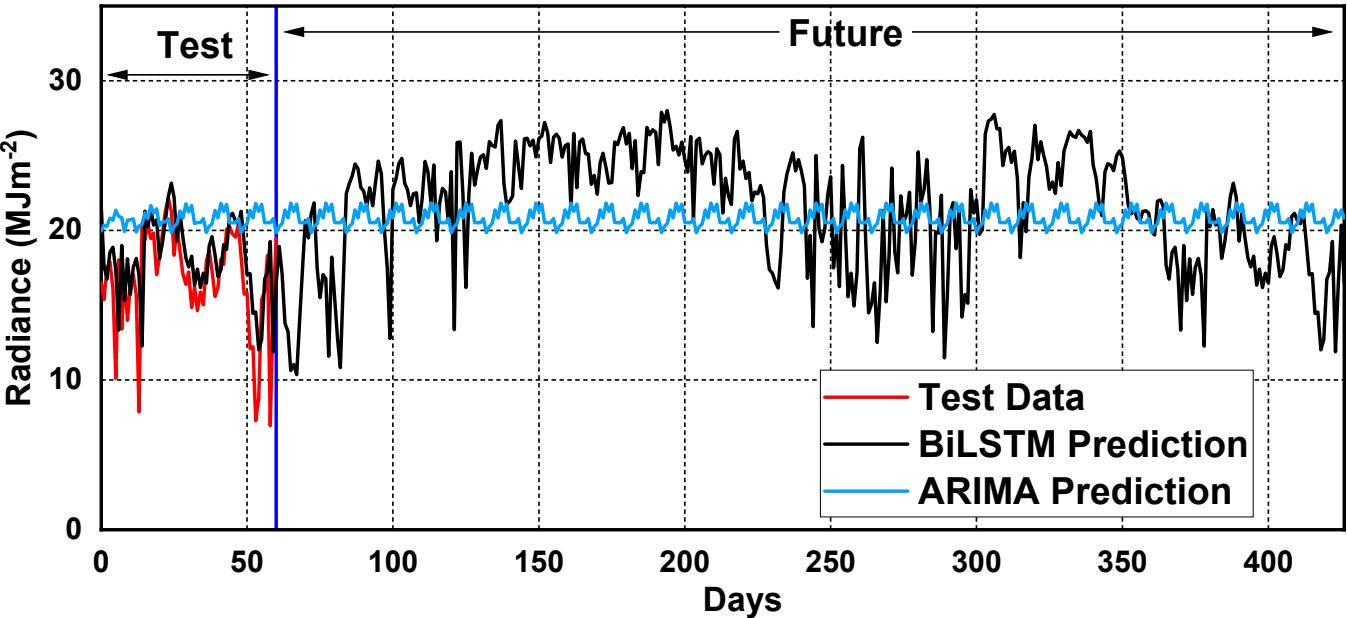

**Figure 16.** Graphical visualization of the comparison between tested and predicted data and one-year prediction of grid daily generation of a solar plant.

Figure 15 exhibits the "daily generation (kWh)" parameter's range of 100,000 to 500,000 units on the *y*-axis, with the number of days on the *x*-axis. The comparison of predicted data using ARIMA (auto-regressive integrated moving average) and Bi-LSTM (bidirectional long short-term memory) and the remaining actual 60-day test data shows that Bi-LSTM's prediction is almost synchronized with the test results, with minor deviations at certain points. However, ARIMA's results are mostly incompatible. After this comparison, the daily generation data are manipulated for the next 12 months (365 days) using both trained prediction models. It is evident in Figure 15 that Bi-LSTM (bidirectional long short-term memory) shows a decrease initially and then achieves a continuous pattern up to 160 days. After the 60th day, ARIMA's prediction is observed to be constantly increasing.

Figure 16 displays the "radiance (MJ/m$^{-2}$)" parameter's range from 0 to 30 on the *y*-axis, with the number of days on the *x*-axis. The comparison of predicted data using ARIMA (auto-regressive integrated moving average) and Bi-LSTM (bidirectional long short-term memory) and the remaining actual 60-day test data shows that Bi-LSTM's prediction is almost synchronized with the test results, but ARIMA's results are incompatible. After this comparison, the radiance data are manipulated for the next 12 months (365 days) using both trained prediction models. Bi-LSTM (bidirectional long short-term memory) exhibits a decrease initially up to the 80th day and then achieves a continuous pattern up to the 220th day. Afterward, an up-and-down pattern is observed. In contrast, the ARIMA (auto-regressive integrated moving average) model remained continuous from start to end.

## 5. Software and System Details

This research utilized a specific set of hardware and software setups for its experimentation. The hardware components used included an Intel(R) Core (TM) i7-10875H CPU running at a base frequency of 2.30 GHz, an NVIDIA GeForce RTX 2060 graphics card with 6 GB of GUP memory, and 16 GB of RAM. The system operated on a 64-bit version of the Windows operating system. The software used for the research included Python 3.7

programming language, along with Keras and TensorFlow version 2.3.1 libraries for deep learning modeling. The selection of these hardware and software setups highlights their significance in ensuring the efficiency and accuracy of the research.

## 6. Conclusions

The results of this study indicate that the Bi-LSTM (bidirectional long short-term memory) model outperformed the ARIMA (auto-regressive integrated moving average) model in terms of accuracy and performance metrics for all three parameters. The Bi-LSTM model's ability to accurately capture the intricate relationships and patterns hidden in the data, which relies on non-linear relationships, was found to be superior to the ARIMA model's reliance on linear relationships. Specifically, for the grid-connected power generation parameter, the Bi-LSTM model achieved a mean absolute error (MAE) of 0.012, while the ARIMA model had an MAE of 0.016. Similarly, for the daily generation parameter, the Bi-LSTM model produced an MAE of 0.019, whereas the ARIMA model managed an MAE of 0.024. Lastly, for the radiance parameter, the Bi-LSTM model had an MAE of 0.027, while the ARIMA model's MAE was 0.032. These results demonstrate that the Bi-LSTM model can offer more accurate and reliable predictions for these parameters, which are crucial for managing solar power facilities effectively. In future research, it would be valuable to investigate the effectiveness of other deep-learning models, such as convolutional neural networks (CNNs) and transformer-based models, for time series forecasting in renewable energy. Additionally, incorporating more relevant features, such as weather data and grid load information, could improve the accuracy of the models' predictions.

**Author Contributions:** Conceptualization, Y.C. and D.X.; methodology, M.S.B.; software, H.N.; validation, M.S.B., M.A. and H.N.; formal analysis, F.M.A.; investigation, Y.C.; resources, D.X.; data curation, Y.C.; writing—original draft preparation, M.S.B.; writing—review and editing, Y.C.; visualization, H.N.; supervision, M.F.; project administration, Y.C.; funding acquisition, M.S.B. All authors have read and agreed to the published version of the manuscript.

**Funding:** This work was supported by the Department of Education of Guangxi Autonomous Region under grant number 2023KY0826.

**Institutional Review Board Statement:** Not applicable.

**Informed Consent Statement:** Not applicable.

**Data Availability Statement:** The corresponding author can provide access to the data upon request, which was analyzed for this research.

**Acknowledgments:** The authors are highly grateful to their affiliated universities and institutes for providing research facilities.

**Conflicts of Interest:** The authors declare no conflict of interest.

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
