# Peer review of "Evaluation of Machine Learning Models for Smart Grid Parameters: Performance Analysis of ARIMA and Bi-LSTM"

_sustainability, doi:10.3390/su15118555_

Round 1

Reviewer 1 Report

Comments

The paper studied the Integrating Renewable Energy Forecasting with Artificial Intelligence: A Study on the Use of Machine Learning Techniques for Smart Grid Optimization. It seems a hot topic, as many research papers have been published in this area. Below, I have provided my comments, which are as follows:

It is not clear to me what the difference is between your study and the published study

doi: 1. 10.1109/ICETECC56662.2022.10069570.

1.       doi: 10.3390/s22239314.

2.       doi: 10.1109/ICPESG.2018.8384493.

3.       doi: 10.1109/FIT.2016.067.

4.       doi: 10.3390/en15228333.

5.       doi: 10.1109/FIT57066.2022.00014.

* What is the main contribution of your previous study to the current study? I need clarification on it.

* There are many grammatical issues in the paper.

* When you talk about published studies, you should provide references. See this sentence in your paper that you claimed, ". For a large number of users,".

* The introduction and literature review needs to be comprehensively expanded, which confuses the author to follow you on what you plan to do with your paper.

* The literature review did not cover all studies close to your research area, which is a weakness of your study. Please summarize it in a table by adding more recent and relevant papers.

* I need clarification on why the author just provided the flowchart of the ARIMA and Bi-LSTM and then, instead of providing and explaining the initialization part, just provided some examples of a different model. Also, each part of Figure 1, should be explained in detail to the reader how to utilize the Bi-Lstm. It is unclear to me.

The abstract and conclusion must be revised by including rationale, solid reasoning, solutions, etc.

In this round of revision, I will go for a significant revision to see how the author responds to my comments.

Title is too long, it must be within 15 words. Revise the title.

Minor grammatical and typos check 

Author Response

Reviewer 1

  1. The paper studied the Integrating Renewable Energy Forecasting with Artificial Intelligence: A Study on the Use of Machine Learning Techniques for Smart Grid Optimization. It seems a hot topic, as many research papers have been published in this area. Below, I have provided my comments, which are as follows:

It is not clear to me what the difference is between your study and the published study

doi: 1. 10.1109/ICETECC56662.2022.10069570.

  1. doi: 10.3390/s22239314.
  2. doi: 10.1109/ICPESG.2018.8384493.
  3. doi: 10.1109/FIT.2016.067.
  4. doi: 10.3390/en15228333.
  5. doi: 10.1109/FIT57066.2022.00014.

Response

Respected reviewer, these articles likely discuss the integration of renewable energy resources into smart grids and the efficient management and control of power production, which are mentioned in the paper as the challenges of the fourth energy revolution. These articles may also cover the use of artificial intelligence (AI) techniques, such as machine learning, in addressing these challenges and improving energy production control and management. The articles may provide insights into how IoT applications and monitoring can be utilized in smart grids to optimize the integration of renewable energy resources, leading to more efficient and sustainable power systems, as mentioned in the text. Whereas in our study, a prediction work has been done on a real time data parameters of a solar power plant through machine learning statistical model ARIMA and Bi-LSTM and a comparison is made. As both of these models gave the best results of prediction but Bi-LSTM outperformed. The only similarity of the above-mentioned articles and this study is the use of machine learning models to get the final results and visualization.

  1. What is the main contribution of your previous study to the current study? I need clarification on it.

Response

Respected reviewer, in many previous studies it has been observed that only single, hybrid and ensemble models are used for the prediction of energy plants parameters. Mostly authors used imaginary data to achieve the results but, in this case, the one-year real time data is gathered from a large-scale solar power plant. Also, the models used in this study have been explored with various hyperparameters tuning and the best possible results are presented in this study. The data consists of three important influential parameters which has been kept under observation by using different statistical/machine learning models to acquire the results but results were not up to the mark. Only Auto Regressive Integrated Moving Average (ARIMA) in statistical approach and Bi-LSTM (Long Short-Term Memory) in machine learning gave the best performance on the data. Then by comparing the results obtained by these two models, it has been observed that Bi-LSTM’s performance is up to the mark and prediction is accurate. So that is why in our current study, we choose these two models. In near future for the further improvement, these models will be combined with other suitable statistical and deep learning models.      

  1. There are many grammatical issues in the paper.

Response

Respected reviewer, thanks for noticing, grammatical mistakes has been removed and highlighted.

  1. When you talk about published studies, you should provide references. See this sentence in your paper that you claimed, ". For a large number of users,".

Response

The introduction section has been modified completely and all references have been provided. I hope it will fulfill the requirements.

  1. The introduction and literature review need to be comprehensively expanded, which confuses the author to follow you on what you plan to do with your paper.

Response

Respected reviewer, introduction has been changed completely and tried to expand comprehensively as per suggestion. Hope you will be satisfied with it now.

  1. The literature review did not cover all studies close to your research area, which is a weakness of your study. Please summarize it in a table by adding more recent and relevant papers.

Response

Respected reviewer, I am thankful to you for pointing this major deficiency. I read it thoroughly and feel that it really needs to improve more. Now I have cited more research articles relevant to this research as per your suggestion and highlighted too.

  1. I need clarification on why the author just provided the flowchart of the ARIMA and Bi-LSTM and then, instead of providing and explaining the initialization part, just provided some examples of a different model. Also, each part of Figure 1, should be explained in detail to the reader how to utilize the Bi-LSTM. It is unclear to me.

Response

Respected reviewer, initialization part and each step of flowchart of ARIMA and Bi-LSTM has been explained as per suggestion. I am thankful to you to point out this mistake and also in figure 1. Each part of the flow chart has been discussed now and highlighted too.

  1. The abstract and conclusion must be revised by including rationale, solid reasoning, solutions, etc.

Response

Respected reviewer, abstract and conclusion has been revised according to your given direction and as per my approach. Now it is more reasonable meaningful and purposeful. Also highlighted the changes. 

  1. Title is too long; it must be within 15 words. Revise the title.

Response

The title has been revised.  

  1. In this round of revision, I will go for a significant revision to see how the author responds to my comments.

Response

Respected reviewer, first of all I am very much thankful to you for pointing all the deficiencies, flaws and mistakes out. Moreover, all of your suggestions and comments has been answered with full attention and sincerity up to my level best. Hopefully you will be satisfied with all these explanations.

Reviewer 2 Report

To start with, I would like to thank the authors for their work in terms of used language and interesting topic

The paper under review is devoted to the application ARIMA and Bi-LSTM machine learning techniques for predicting solar power production. Mean absolute error (MAE) and root mean squared error (RMSE) are used for assessing the quality of trained models.

My remarks on the paper strengths are:

·         The theme of the article might be considered in the SI journal topic “Energy Technologies, Challenges and Solutions for a Sustainable (Energy) World

·         The structure of paper is good with all essential sections required for scientific papers

·         Content is more or less coherent and cohesive.

·         Used English is at high level.

·         Reference are sufficient and up-to-date

However, the paper has drawbacks:

1.     The title does not reflect the essence of the paper. Where is “Integrating Renewable Energy Forecasting “ to?  “Renewable Energy Forecasting”  - maybe energy production ? “A Study on the Use of Machine Learning Techniques for Smart Grid Optimization”  - nope. There are only  ARIMA and Bi-LSTM techniques, do not speak for all the techniques if you don't consider them.

2.     Line 44-45 . I do not recommend to start with ” This research also focuses on one year of historical solar power production data and evaluates the performance parameters of the ARIMA and Bi-LSTM models in predicting solar power production for the coming next year. It should be un the end of Introduction. In addition, then you immediately discuss GRU, LSTM and RNN, but it doesn’t correspond to the research.

3.     Then up to line 136, there are lots of references, but it is not clear why did you do? What are the outcomes of the conducted  literature review?

4.     Where is the relevance of the study? Authors have not justified the using ARIMA and Bi-LSTM ML approaches. Please, work on the introduction section better, make it more coherent and cohessive.

5.     What is the date taken for forecasting? Region, company and so on. How was data ontained.

6.     Figure 1 represents a flowchart but it is performed in not correct from. Please, refer to ISO 5807:1985 (or at least, start with  https://en.wikipedia.org/wiki/Flowchart) .Where are Input/Output blocks? (you have it , but do not show)

7.     Where is the flowchart in Figure 3? It look like a flowchart a bit only

8.     Figure 4 .  Where are explanations of what is inside figure ?

9.     Line 394 – 473. Where are outcomes of the section? Where are discussion of data? Correlation and frequency distribution.

10.  Line 480-482. A comprehensive literature review is not shown for readers to judge about the best accuracy for Autoregressive Integrated Moving Average

11.  Line479. Only here abbreviations  ARIMA and Bi-LSTM are defined. Please, difine them as soon as they appear in text.

12.  Please set (a), (b), (c), (d) in figures 8 -10 in order to catch the correct figure with correct caption

13.  Line 614 . Where is ‘Firstly”

14.  The discussion involves comparing metrics for only ARIMA and Bi-LSTM techniques, this is not enough! what about other ML techniques? (refer to concern 10)

15.  English is good but it has typos, the manuscript needs prrofreading

16.  The calculations is made in Phyton? It will be great to see script in additional files especially if jupyter notebook was used. Please, mention it in Methodology.

17.  4 impropriate self-citations were detected [1-4]

 English is good but it has typos, the manuscript needs prrofreading

Author Response

Reviewer 2.

  1. To start with, I would like to thank the authors for their work in terms of used language and interesting topic

The paper under review is devoted to the application ARIMA and Bi-LSTM machine learning techniques for predicting solar power production. Mean absolute error (MAE) and root mean squared error (RMSE) are used for assessing the quality of trained models.

My remarks on the paper strengths are:

  • The theme of the article might be considered in the SI journal topic “Energy Technologies, Challenges and Solutions for a Sustainable (Energy) World”
  • The structure of paper is good with all essential sections required for scientific papers
  • Content is more or less coherent and cohesive.
  • Used English is at high level.
  • References are sufficient and up-to-date

However, the paper has drawbacks:

Response

I am very much thankful to you for such a nice remark and pointing out all the deficiencies and drawbacks. I am also thankful for your kind suggestions and tried to answer all your comments with proper care and attention.

  1. The title does not reflect the essence of the paper. Where is “Integrating Renewable Energy Forecasting “to?  “Renewable Energy Forecasting” - maybe energy production? “A Study on the Use of Machine Learning Techniques for Smart Grid Optimization” - nope. There are only ARIMA and Bi-LSTM techniques, do not speak for all the techniques if you don't consider them.

Response

Respected reviewer, title has been changed as per your suggestion and tried to make it more similar with the work done. Changes are highlighted.

  1. Line 44-45. I do not recommend to start with” This research also focuses on one year of historical solar power production data and evaluates the performance parameters of the ARIMA and Bi-LSTM models in predicting solar power production for the coming next year. It should be un the end of Introduction. In addition, then you immediately discuss GRU, LSTM and RNN, but it doesn’t correspond to the research.

Response

Respected reviewer thanks to point out this mistake. Mentioned lines has been removed from the text and has been put at the end of introduction portion as per your suggestion.

  1. Then up to line 136, there are lots of references, but it is not clear why did you do? What are the outcomes of the conducted literature review?

Response

Respected reviewer, as per your suggestion references has been minimized. The text now reflect more relevance and exact elaboration now. All the irrelevant references have been removed. Thank you for pointing out this correction to improve the paper quality.

  1. Where is the relevance of the study? Authors have not justified the using ARIMA and Bi-LSTMML approaches. Please, work on the introduction section better, make it more coherent and cohesive.

Response

Respected reviewer, introductory portion has been fully changed and improved as per your instructions. Thanks to point and put forward such a big deficiency in this study.

  1. What is the date taken for forecasting? Region, company and so on. How was data obtained?

Response

The details of the solar power plant, region and company can be found in the reference [7]. The reference has been added and also highlighted in the paper and the data from January 2022 to December 2022 was used.

[7] Abubakar, M., Che, Y., Ivascu, L., Almasoudi, F. M., & Jamil, I. (2022). Performance analysis of energy production of large-scale solar plants based on artificial intelligence (machine learning) technique. Processes, 10(9), 1843. https://doi.org/10.3390/pr10091843

  1. Figure 1 represents a flowchart but it is performed in not correct from. Please, refer to ISO 5807:1985 (or at least, start with https://en.wikipedia.org/wiki/Flowchart). Where are Input/Output blocks? (You have it, but do not show)

Response

Respected reviewer, as per your suggestion, figure 1 has been corrected and input and output blocks has been added in the diagram. Thanks

  1. Where is the flowchart in Figure 3? It looks like a flowchart a bit only

Response

Respected reviewer, Figure 3. Is actually not a flowchart, it is the demonstration of the ARIMA model process which is discussed in previous paragraph. However, some improvements in the discussion have been done and highlighted.

  1. Figure 4.  Where are explanations of what is inside figure?

Response

Respected reviewer, as per your suggestion, explanation has been added in the figure and also discussed in the text and highlighted too.

  1. Line 394 – 473. Where are outcomes of the section? Where are discussion of data? Correlation and frequency distribution.

Response

Respected reviewer, line 394 to onwards, discussion has been added to the text and highlighted.

  1. Line 480-482. A comprehensive literature review is not shown for readers to judge about the best accuracy for Autoregressive Integrated Moving Average

Response

Respected reviewer, Literature review has been updated as per your mentioned mistake. Now both the models have been discussed in literature review thoroughly.

  1. Line479. Only here abbreviations ARIMA and Bi-LSTM are defined. Please, define them as soon as they appear in text.

Response

Respected reviewer, mentioned mistake has been removed even in all text, abbreviations are fully defined now.

  1. Please set (a), (b), (c), (d) in figures 8 -10 in order to catch the correct figure with correct caption.

Response

Respected reviewer, mentioned mistake has been removed and (a), (b), (c), (d) are added.

  1. Line 614. Where is ‘Firstly”

Response

Respected reviewer, 487 line is showing the first part in which ARIMA model results are discussed.

  1. The discussion involves comparing metrics for only ARIMA and Bi-LSTM techniques, this is not enough! what about other ML techniques? (Refer to concern 10)

Response

Respected reviewer, other ML techniques are now discussed along with ARIMA and BI-LSTM models in the introduction portion and highlighted too. Thanks for such a good suggestion to improve the quality of the work.

  1. English is good but it has typos, the manuscript needs proofreading.

Response

Proofreading has been done.

  1. The calculations is made in Phyton? It will be great to see script in additional files especially if jupyter notebook was used. Please, mention it in Methodology.

Response

Respected reviewer, yes, the calculation has been done in python and the concerned coding file is attached in additional files for your observation.

  1. 4 impropriate self-citations were detected [1-4].

Response

These have been removed.

Reviewer 3 Report

This paper discusses the Bi-LSTM model's superior accuracy and performance metrics over the ARIMA model across all three parameters highlight its effectiveness in capturing complex patterns and dependencies in the data. Overall, this research paper offers valuable insights into the advancement of smart grids and the potential of machine learning in renewable energy management, making it an interesting and significant contribution to the field.

Before final submission, kindly address the following comments in the revised paper.

1. Mention the software platform and the system specifications on which these simulations are executed in the results and discussion section. It can be interesting for the readers.

2. In lines from 134-136, discuss more about Bi-LSTM and ARIMA models from few more updated references to make the introduction more interesting.

3. At some places in the text, the S.I. units are missing for the parameters (power, radiance, daily generation). Kindly, check it and give proper S.I. units to the parameters.

4. Suggest some future work your research in the results and discussion section.

5. Also check for English mistakes and correct it carefully.

This paper discusses the Bi-LSTM model's superior accuracy and performance metrics over the ARIMA model across all three parameters highlight its effectiveness in capturing complex patterns and dependencies in the data. Overall, this research paper offers valuable insights into the advancement of smart grids and the potential of machine learning in renewable energy management, making it an interesting and significant contribution to the field.

Before final submission, kindly address the following comments in the revised paper.

1. Mention the software platform and the system specifications on which these simulations are executed in the results and discussion section. It can be interesting for the readers.

2. In lines from 134-136, discuss more about Bi-LSTM and ARIMA models from few more updated references to make the introduction more interesting.

3. At some places in the text, the S.I. units are missing for the parameters (power, radiance, daily generation). Kindly, check it and give proper S.I. units to the parameters.

4. Suggest some future work your research in the results and discussion section.

5. Also check for English mistakes and correct it carefully.

Author Response

Reviewer 3.

  1. This paper discusses the Bi-LSTM model's superior accuracy and performance metrics over the ARIMA model across all three parameters highlight its effectiveness in capturing complex patterns and dependencies in the data. Overall, this research paper offers valuable insights into the advancement of smart grids and the potential of machine learning in renewable energy management, making it an interesting and significant contribution to the field.

Before final submission, kindly address the following comments in the revised paper.

Response

Thanks for appreciating comments.

  1. Mention the software platform and the system specifications on which these simulations are executed in the results and discussion section. It can be interesting for the readers.

Response

Respected reviewer, software and system details has been added to the text and no doubt it will enhance the interest of the reader. Thanks

  1. In lines from 134-136, discuss more about Bi-LSTM and ARIMA models from few more updated references to make the introduction more interesting.

Response

The more relevant references have been added and revised.

  1. At some places in the text, the S.I. units are missing for the parameters (power, radiance, and daily generation). Kindly, check it and give proper S.I. units to the parameters.

Response

Mentioned mistake has been removed and highlighted.

  1. Suggest some future work your research in the results and discussion section.

Response

Respected reviewer, future work and ideas has been discussed as per suggestion and highlighted.

  1. Also check for English mistakes and correct it carefully.

Response

Respected reviewer, proof reading has been done and mistakes has been corrected and highlighted.

Author Response

Reviewer 4.

  1. The proposed work was intended to investigate “Integrating Renewable Energy Forecasting with Artificial Intelligence: A Study on the Use of Machine Learning Techniques for Smart Grid Optimization”. However, minor revision is required considering the following points mentioned below:

Response

Thanks for appreciating comments.

  1. Provide the proper references to all the mathematical equations mentioned in the paper.

Response

Respected reviewer, references have been put in the text as per instruction.

  1. There are grammatical and spelling mistakes in the paper, carefully review it and address them in the revised version.

Response

Respected reviewer, mistakes has been removed from the text as per your instructions. Also, paper has been checked thoroughly.

  1. The paper lacks the citations of year 2022 and 2023, kindly cite some more latest work from the year 2022 and 2023.

Response

Respected reviewer, updated and latest citations has been put in the text.

  1. Quality of figures is so important too. Please provide some high-resolution figures (figure 8, 9 & 10).

Response

The quality has been improved in the revised version.

  1. Future recommendations should be added in the conclusion.

Response

As per your instructions, future ideas and work has been added to the study also in conclusion section. Thanks

Round 2

Reviewer 1 Report

The following comments raised in the earlier round have not been properly addressed. Authors are advised to address the comments and resubmit the manuscript for consideration. 

Comments

The paper studied the Integrating Renewable Energy Forecasting with Artificial Intelligence: A Study on the Use of Machine Learning Techniques for Smart Grid Optimization. It seems a hot topic, as many research papers have been published in this area. Below, I have provided my comments, which are as follows:

It is not clear to me what the difference is between your study and the published study

doi: 1. 10.1109/ICETECC56662.2022.10069570.

1.       doi: 10.3390/s22239314.

2.       doi: 10.1109/ICPESG.2018.8384493.

3.       doi: 10.1109/FIT.2016.067.

4.       doi: 10.3390/en15228333.

5.       doi: 10.1109/FIT57066.2022.00014.

6.       https://doi.org/10.1016/j.apenergy.2023.120640

* What is the main contribution of your previous study to the current study? I need clarification on it.

* There are many grammatical issues in the paper.

* When you talk about published studies, you should provide references. See this sentence in your paper that you claimed, ". For a large number of users,".

* The introduction and literature review needs to be comprehensively expanded, which confuses the author to follow you on what you plan to do with your paper.

* The literature review did not cover all studies close to your research area, which is a weakness of your study. Please summarize it in a table by adding more recent and relevant papers.

* I need clarification on why the author just provided the flowchart of the ARIMA and Bi-LSTM and then, instead of providing and explaining the initialization part, just provided some examples of a different model. Also, each part of Figure 1, should be explained in detail to the reader how to utilize the Bi-Lstm. It is unclear to me.

The abstract and conclusion must be revised using rationale, solid reasoning, solutions, etc.

In this round of revision, I will go for a significant revision to see how the author responds to my comments.

The title is too long; it must be within 15 words. Revise the title.

Typos and minor grammar mistakes

Author Response

Reviewer 1

  1. The paper studied the Integrating Renewable Energy Forecasting with Artificial Intelligence: A Study on the Use of Machine Learning Techniques for Smart Grid Optimization. It seems a hot topic, as many research papers have been published in this area. Below, I have provided my comments, which are as follows:

It is not clear to me what the difference is between your study and the published study

  1. doi: 10.1109/ICETECC56662.2022.10069570.
  2. doi: 10.3390/s22239314.
  3. doi: 10.1109/ICPESG.2018.8384493.
  4. doi: 10.1109/FIT.2016.067.
  5. doi: 10.3390/en15228333.
  6. doi: 10.1109/FIT57066.2022.00014.

Response

Respected Reviewer, after a through consultation, the clarification about the differences between our study and the mentioned published studies are given as follows:

  1. In doi:10.1109/ICETECC56662.2022.10069570 the importance of reliability and controls to prevent power system outages and the potential for Smart Grids to provide cost-effective solutions to optimize load demand, manage power sources and stabilize equipment operations are discussed. However, due to the early stages of Smart Grid (SG) development, the text explains that SGs are vulnerable to disturbances caused by changes in demand, grid disruptions, and renewable energy variations. Also the article provides an overview of different types of power system disturbances and their impacts on SG stability, as well as an analysis of common linear and nonlinear control strategies used in SG systems. Finally, the text emphasizes the need for more robust operational and control approaches to enhance SG stability. Hence the core difference between the two researches is that the first one discusses the importance of reliability and controls for enhancing the stability of smart grids, while the our research focuses on the application of machine learning techniques to optimize the integration of renewable energy resources into smart grids for improving energy production control and management.
  2. In the same way, the main difference between doi: 10.3390/s22239314 and out study is that the first one discusses the development of a hybrid solution for DFIG wind turbines with FRT capabilities to address concerns about the reliability and stability of power systems caused by large-scale wind power integration, while the second one investigates the application of machine learning techniques, specifically ARIMA and Bi-LSTM models, for predicting solar power production to optimize the integration of renewable energy resources into smart grids, leading to more efficient and sustainable power systems.
  3. In doi: 10.1109/ICPESG.2018.8384493 discusses the development of a hybrid solution for wind turbines to enhance reliability and stability during grid failures. It uses a modified switch-type fault current limiter and a DC chopper to keep the rotor current and DC-link voltage within limits. Simulation tests show that the proposed scheme outperforms existing control approaches. Whereas in our study the use of machine learning techniques, specifically ARIMA and Bi-LSTM models, to predict solar power production for the next year. The study finds that the Bi-LSTM model outperforms the ARIMA model in terms of accuracy and can successfully identify patterns and relationships in real-time data. The findings suggest that machine learning can optimize renewable energy integration into smart grids.
  4. Similarly in doi: 10.1109/FIT.2016.067 discusses control methods for a Doubly Fed Induction Generator (DFIG) tied to the grid, focusing on decoupling the stator currents for grid voltage support and reactive power injection during grid faults while in our research, the application of machine learning techniques, specifically ARIMA which is a statistical approach and deep learning Bi-LSTM models, for predicting solar power production in the context of integrating renewable energy resources into smart grids are discussed.
  5. The main difference between doi: 10.3390/en15228333 and our study is that first one is only discussing about Pakistan's energy crisis and the potential for wind energy to be a viable solution in four specific zones in the Sindh province while in research focuses on the integration of renewable energy resources into smart grids, specifically the application of machine learning techniques for predicting solar power production.
  6. Lastly the main difference between the two studies is that doi: 10.1109/FIT57066.2022.00014 discusses the use of renewable energy resources such as wind generation, photovoltaic facilities, fuel cells, and batteries in distribution systems, while also emphasizing the advantages of electric loads and controllable loads. It also describes the modeling and simulation of solar power modules through MATLAB. On the other hand, our research focuses on the integration of renewable energy resources into smart grids, particularly through the application of machine learning techniques such as ARIMA and Bi-LSTM models to predict solar power production for the next year. The paragraph highlights the benefits of machine learning in optimizing the integration of renewable energy resources into smart grids, leading to more efficient and sustainable power systems.
  7. What is the main contribution of your previous study to the current study? I need clarification on it.

Response

Dear reviewer, previous studies have mainly used single, hybrid, and ensemble models for predicting energy plant parameters, often relying on simulated data. I have gone through many research articles and found that no one used ARIMA model which is statistical approach along with a deep learning model for a solar power plant’s parameter. In contrast, our study uses real-time data collected over one year from a large-scale solar power plant. We also explore various hyperparameters for the models used in this study to obtain the best possible results. Our dataset includes three important parameters, which we analyze using different statistical and machine learning models. However, only the Auto Regressive Integrated Moving Average (ARIMA) statistical model and the Bi-LSTM (Long Short-Term Memory) machine learning model demonstrated the best performance on the data. After comparing the results obtained by these two models, we found that Bi-LSTM was the most accurate. Therefore, we chose to use these two models in our current study. In the future, we plan to combine these models with other suitable statistical and deep learning models to further improve our predictions. Hope the response would be satisfactory. Thanks.

  1. There are many grammatical issues in the paper.

Response

Respected reviewer, thanks for noticing, grammatical mistakes has been removed and highlighted.

  1. When you talk about published studies, you should provide references. See this sentence in your paper that you claimed, ". For a large number of users,".

Response

The introduction section has been modified completely and all references have been provided. But as you mentioned in your response about my claim “For a large number of users”, I am unable to find such wordings in my current study. So it is a humble request to you that please it may kindly be ignored. Thanks

  1. The introduction and literature review need to be comprehensively expanded, which confuses the author to follow you on what you plan to do with your paper.

Response

Respected reviewer, introduction/literature review has been changed completely and tried to expand comprehensively as per your suggestion. More references has been added and many new ideas are discussed in it. Hope you will be satisfied with it now.

  1. The literature review did not cover all studies close to your research area, which is a weakness of your study. Please summarize it in a table by adding more recent and relevant papers.

Response

Respected reviewer, I am thankful to you for pointing this major deficiency. I read it thoroughly and felt that it really needs to be improved. Now more references has been added relevant to this research as per your suggestion and highlighted too.

  1. I need clarification on why the author just provided the flowchart of the ARIMA and Bi-LSTM and then, instead of providing and explaining the initialization part, just provided some examples of a different model. Also, each part of Figure 1, should be explained in detail to the reader how to utilize the Bi-LSTM. It is unclear to me.

Response

Dear reviewer, I have incorporated your suggestions and provided a detailed explanation of the initialization part and each step in the flowchart of both ARIMA and Bi-LSTM models. I appreciate you for bringing this to my attention and for pointing out the mistake in Figure 1. I have now thoroughly discussed and highlighted each part of the flowchart. Thank you for your valuable feedback.

  1. The abstract and conclusion must be revised by including rationale, solid reasoning, solutions, etc.

Response

Dear respected reviewer,

I would like to inform you that I have made revisions to the abstract and conclusion of the paper as per your guidance and incorporating my approach as well. I have carefully considered your suggestions and have worked to make the abstract and conclusion more reasonable, meaningful, and purposeful.

Furthermore, I have highlighted the changes made in both sections so that they are easily identifiable. I hope that the revised abstract and conclusion meet your expectations, and I appreciate your valuable feedback and guidance throughout the review process.

  1. Title is too long; it must be within 15 words. Revise the title.

Response

The title has been revised.Thanks  

  1. In this round of revision, I will go for a significant revision to see how the author responds to my comments.

Response

Dear respected reviewer,

I would like to express my sincere gratitude for taking the time to thoroughly review my work and provide constructive feedback. Your insightful comments have been invaluable in identifying the deficiencies, flaws, and mistakes in my work. I want to assure you that I have taken your suggestions and comments seriously, and have made significant revisions to my work. I have given full attention and sincerity to addressing each and every one of your concerns to the best of my ability. I truly hope that my explanations and revisions will meet your expectations and be satisfactory. Once again, I appreciate your effort and expertise in providing feedback that has helped me improve my work.

Thank you for your time and consideration.

Reviewer 2 Report

Dear authors, thank you for your revision and addressing my concerns. Good study. Go on.

English is good, no serious problems

Author Response

Thanks for your appreciating comments and useful suggestions. 

Round 3

Reviewer 1 Report

Many thanks to the authors for addressing my comments. However, still, a few comments are not addressed properly. For example,

The comparison with existing literature is only included in the response file and not in the revised manuscript. Authors are supposed to add compassion in both response and manuscript. 

Minor

Author Response

1. Many thanks to the authors for addressing my comments.

Response:

Thanks for the appreciation.

2. However, still, a few comments are not addressed properly. For example, the comparison with existing literature is only included in the response file and not in the revised manuscript. Authors are supposed to add compassion in both response and manuscript.

Response:

The comparison with existing literature review is included in the revised manuscript and also highlighted.

Round 4

Reviewer 1 Report

My comments are properly addressed. I have no further comments.